# Single-cell transcriptomics and epigenomics unravel the role of monocytes in neuro-blastoma bone marrow metastasis

Irfete S. Fetahu ®[1,10] ✉, Wolfgang Esser-Skala[2,10], Rohit Dnyansagar[2,10], Samuel Sindelar[2], Fikret Rifatbegovic ®[1], Andrea Bileck ®[3,4], Lukas Skos ®[3], Eva Bozsaky ®[1], Daria Lazic[1], Lisa Shaw ®[5], Marcus Tötzl ®[1], Dora Tarlungeanu[1], Marie Bernkopf ®[1], Magdalena Rados[1], Wolfgang Weninger[5], Eleni M. Tomazou ®[1], Christoph Bock ®[6,7], Christopher Gerner ®[3,4], Ruth Ladenstein ®[8,9], Matthias Farlik ®[5], Nikolaus Fortelny ®[2,11] ✉ & Sabine Taschner-Mandl ®[1,11] ✉

Metastasis is the major cause of cancer-related deaths. Neuroblastoma (NB), a childhood tumor has been molecularly defined at the primary cancer site, however, the bone marrow (BM) as the metastatic niche of NB is poorly characterized. Here we perform single-cell transcriptomic and epigenomic profiling of BM aspirates from 11 subjects spanning three major NB subtypes and compare these to five age-matched and metastasis-free BM, followed by in-depth single cell analyses of tissue diversity and cell-cell interactions, as well as functional validation. We show that cellular plasticity of NB tumor cells is conserved upon metastasis and tumor cell type composition is NB subtype-dependent. NB cells signal to the BM microenvironment, rewiring via macro-phage mgration inhibitory factor and midkine signaling specifically mono-cytes, which exhibit M1 and M2 features, are marked by activation of pro- and anti-inflammatory programs, and express tumor-promoting factors, reminiscent of tumor-associated macrophages. The interactions and pathways characterized in our study provide the basis for therapeutic approaches that target tumor-to-microenvironment interactions.

Neuroblastoma (NB) accounts for 15% of childhood cancer-related deaths, where >90% of metastatic stage (stage M) NB tumors disseminate to the bone marrow (BM), which acts as a site for disease relapse and progression[1–4]. Genetic NB tumor heterogeneity and plasticity have been suggested to contribute to differentiation or metastasis and relapse, serving as intrinsic oncogenic drivers[5–8]. Main genetic factors involved in disease onset and progression include amplification of *MYCN* (MNA), mutation of *TP53*, amplification or

[1]St. Anna Children's Cancer Research Institute, Vienna, Austria. [2]Department of Biosciences and Medical Biology, University of Salzburg, Salzburg, Austria. [3]University of Vienna, Department of Analytical Chemistry, Faculty of Chemistry, Vienna, Austria. [4]Joint Metabolomics Facility, University of Vienna and Medical University of Vienna, Vienna, Austria. [5]Medical University of Vienna, Department of Dermatology, Vienna, Austria. [6]CeMM Research Center for Molecular Medicine of the Austrian Academy of Sciences, Vienna, Austria. [7]Medical University of Vienna, Institute of Artificial Intelligence, Center for Medical Data Science, Vienna, Austria. [8]St. Anna Children's Hospital and St. Anna Children's Cancer Research Institute, Department of Studies and Statistics for Integrated Research and Projects, Vienna, Austria. [9]Medical University of Vienna, Department of Pediatrics, Vienna, Austria. [10]These authors contributed equally: Irfete S. Fetahu, Wolfgang Esser-Skala, Rohit Dnyansagar. [11]These authors jointly supervised this work: Nikolaus Fortelny, Sabine Taschner-Mandl. ✉e-mail: irfete.fetahu@ccri.at; nikolaus.fortelny@plus.ac.at; sabine.taschner@ccri.at

mutation of *ALK* and other *Ras/MAPK* pathway genes, and dysregulation of telomere maintenance via rearrangements of *TERT* or alternative lengthening of telomeres (ALT), which is often associated with mutated or truncated *ATRX* (*ATRX*mut)[9–14]. However, recent whole-genome sequencing studies have identified a scarcity of recurrent somatic alterations[15], but show that a subgroup of metastatic NB is rather defined by large segmental chromosomal aberrations[16] (herein referred to as sporadic).

Numerous studies in recent years[17–23] have focused in defining cell types and lineage trajectories of the developing adrenal medulla, aiming to uncover the cell of origin in NB, which arises from neural crest (NC)-derived sympatho-adrenal progenitor cells at different stages during embryonic development of the sympathetic nervous system[24]. Although the majority of tumor cells in primary NB resemble healthy sympathoblasts, especially in untreated and low-risk tumors[19,21], some tumor cells in pretreated and high-risk cases have shown enriched signatures of chromaffin cells and their progenitors, i.e., Schwann cell precursors (SCP), but also of mesenchymal and NC-like cells, suggesting that their abundance and differentiation state is associated with prognosis[21]. Immune cells within the tumor microenvironment have been shown to carry either tumor promoting or suppressing activities. Studies of primary NB tumors found increased levels of T-, NK, and dendritic cells in the tumor microenvironment of low-risk NB[25–27] compared to high-risk NB[28,29]. Previous studies applying tissue imaging and bulk transcriptomics link myelocytes with inflammatory signatures[30–35]. A comprehensive characterization of the

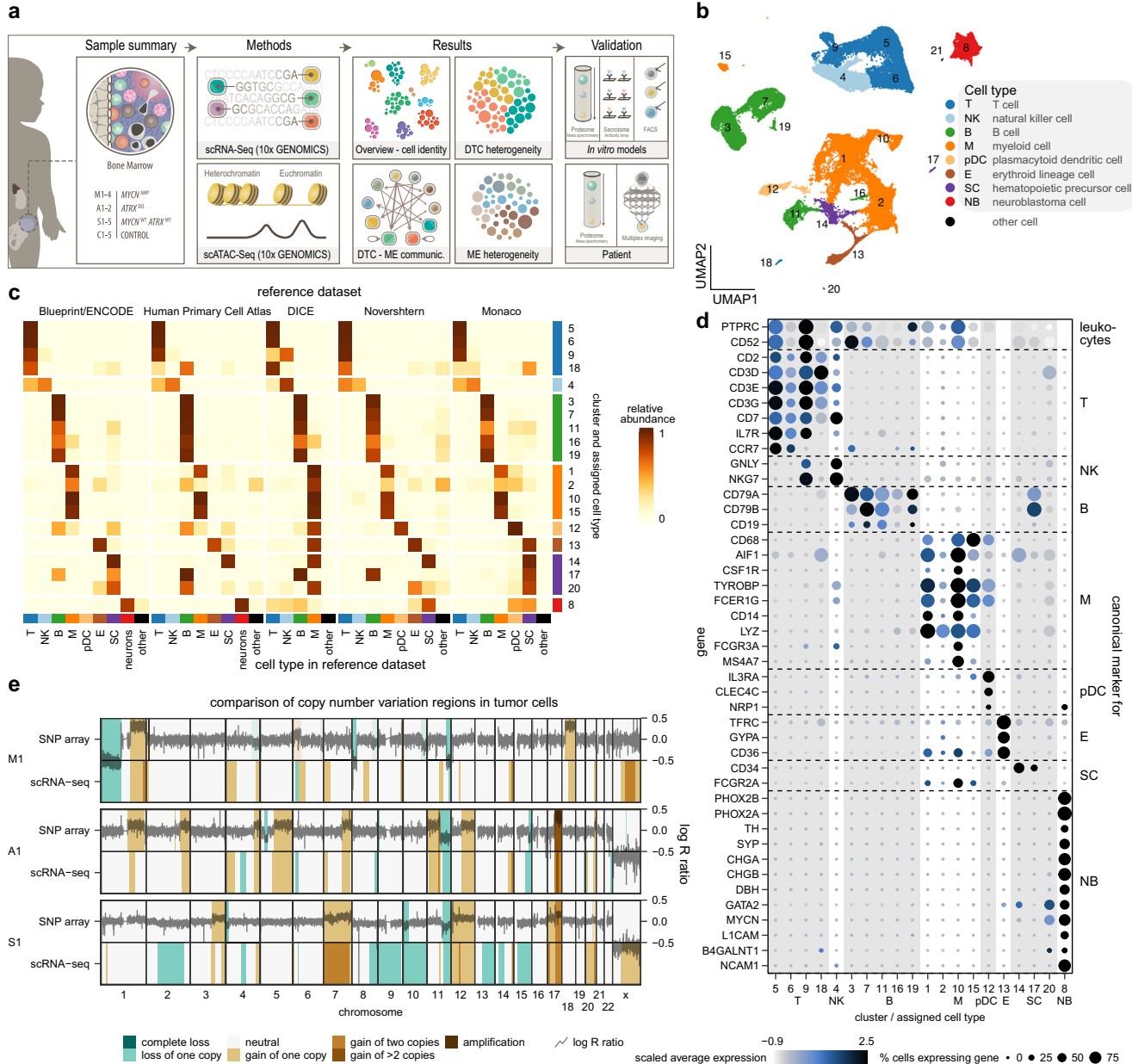

**Fig. 1 | Integrated analysis of NB-infiltrated BM by single-cell RNA sequencing.**
**a** Overview of our experimental and analysis approach. DTC disseminated tumor cells, ME microenvironment **b** UMAP of the overall scRNA-seq dataset (*n* = 80,789 cells) after removal of patient effects. Numbers denote clusters, while colors distinguish cell types. **c** Relative abundances of inferred cell types in each cluster. The annotation to the right indicates the final cell type classification assigned to each cluster. **d** Expression of canonical cell type marker genes in clusters shown in **b**. **e** Comparison of copy number variation regions (colored areas) predicted from scRNA-seq data (bottom rows) to regions deduced from log R ratios (dark lines) in SNP array data (top rows). One patient from each respective NB subtype is shown. Source data are provided as a Source Data file.

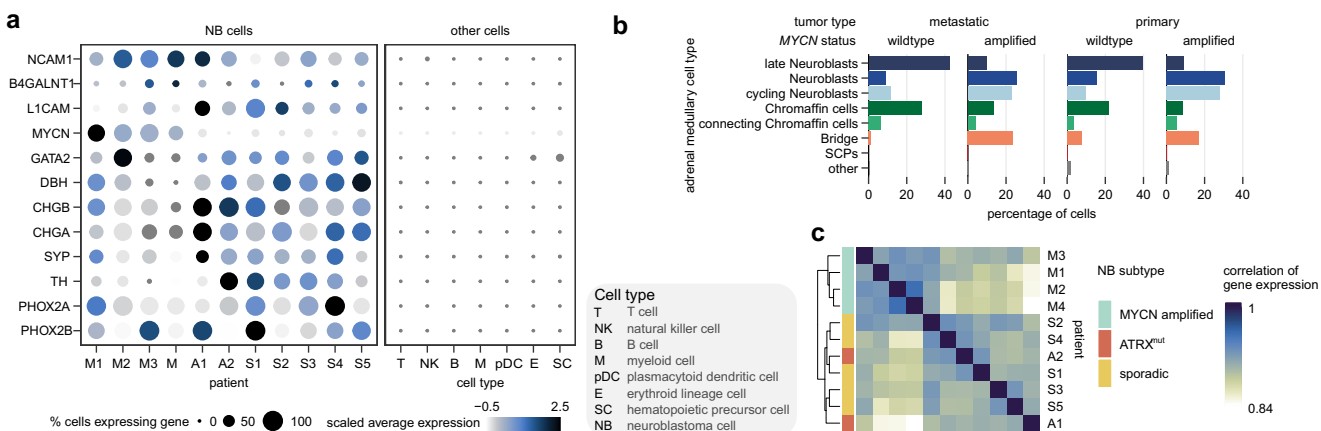

**Fig. 2 | Characterization of the metastatic NB cells in the BM. a** Expression of NB marker genes in NB cells across patients (left) and in other cell types (right). **b** Proportion of NB cells assigned to each adrenal medulla cell type in primary and metastatic tumor cells. NB subtypes are grouped into MNA and non-MNA (A + S) tumors. **c** Correlation analysis of pseudobulk gene expression demarcates the M tumors from A and S tumors. Source data are provided as a Source Data file.

immune microenvironment and crosstalk with tumor cells at the metastatic site remains elusive.

Recently, the use of single-cell technologies have emerged as powerful tools to comprehensively characterize cellular states within healthy and diseased tissues[36]. These approaches have been applied to characterize the tumor heterogeneity, along with the tumor microenvironment, of primary NB tumors[19–21,37], and the composition of adult human BM in normal and disease settings has been investigated, e.g., in leukemia and bone metastases in prostate cancer[38–42]. However, such approaches have yet to be deployed across different NB subgroups, i.e., MNA, *ATRX*[mut], and sporadic at the BM, the metastatic niche of NB. Here, we apply single-cell ATAC-sequencing (scATAC-seq) and single-cell RNA-sequencing (scRNA-seq) across consensus NB subgroups in tandem with proteomics and functional assays to: (i) study differences in cellular plasticity across NB subtypes in metastatic and primary tumors, (ii) investigate interactions between tumor cells and the BM microenvironment, and (iii) unravel metastasis-induced alterations in the BM.

We find that NB subgroups determine cell type composition, and that tumor phenotype is conserved upon metastasis. NB cells primarily interact with myeloid cells, which present with M1 and M2 features, indicated by aberrant pro- and anti-inflammatory core TF regulatory loops, pro-differentiation, and reduction of cell cycle genes as well as expression of tumor-promoting factors. Collectively, these data provide insights into the molecular and cellular architecture of NB across all subgroups, as well as with the potential to inform future studies aimed at improving patient outcomes.

## Results
### The single-cell atlas of neuroblastoma bone marrow metastasis
We integrated genome-wide scRNA-seq and scATAC-seq profiling in BM samples originating from benign tumors (*n* = 5) without BM metastases (ganglioblastoma and ganglioneuroma), herein defined as controls, and 11 samples across metastatic NB subtypes: MNA, *ATRX*[mut], and sporadic (lacking either alteration) (Fig. 1a). Each tumor sample was molecularly and cytogenetically characterized (Supplementary Table 1), substantiating subtype classification. scRNA-seq yielded a total of 80,789 single cells with a median of 1278 genes/cell (Fig. 1b and Supplementary Data 1). Following integration and clustering of the scRNA-seq data (Supplementary Fig. 1a), cells were classified using five reference datasets (Fig. 1c). This yielded seven major cell types, comprised of various types of immune cells: T-cells, NK-cells, B-cells, myeloid cells, and plasmacytoid dendritic cells, followed by erythroid cells and stem cells, which were supported by expression of canonical

marker genes (Fig. 1d)[43–47]. Additionally, we identified a cluster of NB cells, which was (i) classified as neurons, (ii) expressed key NB markers[48,49], and (iii) was absent in control samples (Fig. 1b–d). An unspecified cell cluster consisting of 22 cells was defined as "other" and was excluded from further analysis (Fig. 1b). To further confirm the demarcation between microenvironment and tumor cells, we used two complementary strategies. First, we calculated the tumor infiltration rates based on the evaluation of tumor markers GD2 and L1CAM by flow cytometry (see gating strategy Supplementary Fig. 1b), which were in concordance with the scRNA-seq data assignments (Supplementary Fig. 1c and Supplementary Table 1). Second, we inferred copy number variants (CNVs) based on scRNA-seq data, which were present in NB cells (Supplementary Fig. 1d) and absent in non-NB cells (Supplementary Fig. 1e). The scRNA-seq-based CNV calls were validated using bulk tumor profiling by SNP-arrays (Fig. 1e, Supplementary Data 2), further corroborating our tumor cell assignment.

### The cellular landscape of neuroblastoma cells in the bone marrow metastatic niche
Previous studies in primary tumors have described two transcriptional profiles of NB cells, a committed (nor-) adrenergic and an undifferentiated mesenchymal/neural crest-like (NCC-like) cell type[48,49]. Our data show that metastatic NB cells are primarily defined by a noradrenergic and, to some extent, also an adrenergic signature, albeit the expression of these markers was variable across patients (Fig. 2a, Supplementary Fig. 2a). However, in our sample sets, at the resolution of this study, we did not detect NB cells with a pronounced mesenchymal or NCC-like signature, given that the top 5% of cells with the highest mesenchymal or NCC-like signature score lacked NB-typical marker genes and were present in non-NB cell clusters (Supplementary Fig. 2a–c). Moreover, comparison of metastatic NB cells to adrenal medulla revealed that MNA tumors primarily consist of cells resembling neuroblasts, cycling neuroblasts, and bridge cells. In contrast, non-MNA tumors were characterized by the presence of late neuroblasts and chromaffin cells (Fig. 2b). Leveraging published scRNA-seq data of primary tumors[20], we show that BM metastases display tumor cell phenotypes comparable to the primary site, which were reported in[21] (Fig. 2b, Supplementary Fig. 2d). Correlation analysis of gene expression in primary and metastatic NB showed that MNA tumors cluster together, irrespective of the tumor site. *ATRX*[mut] and sporadic subtypes, form a separate cluster compared to MNA tumors, further substantiating their divergent biology. Moreover, they are marked by transcriptional differences between primary and metastatic tumors (Fig. 2c, Supplementary Fig. 2e). Together, our data

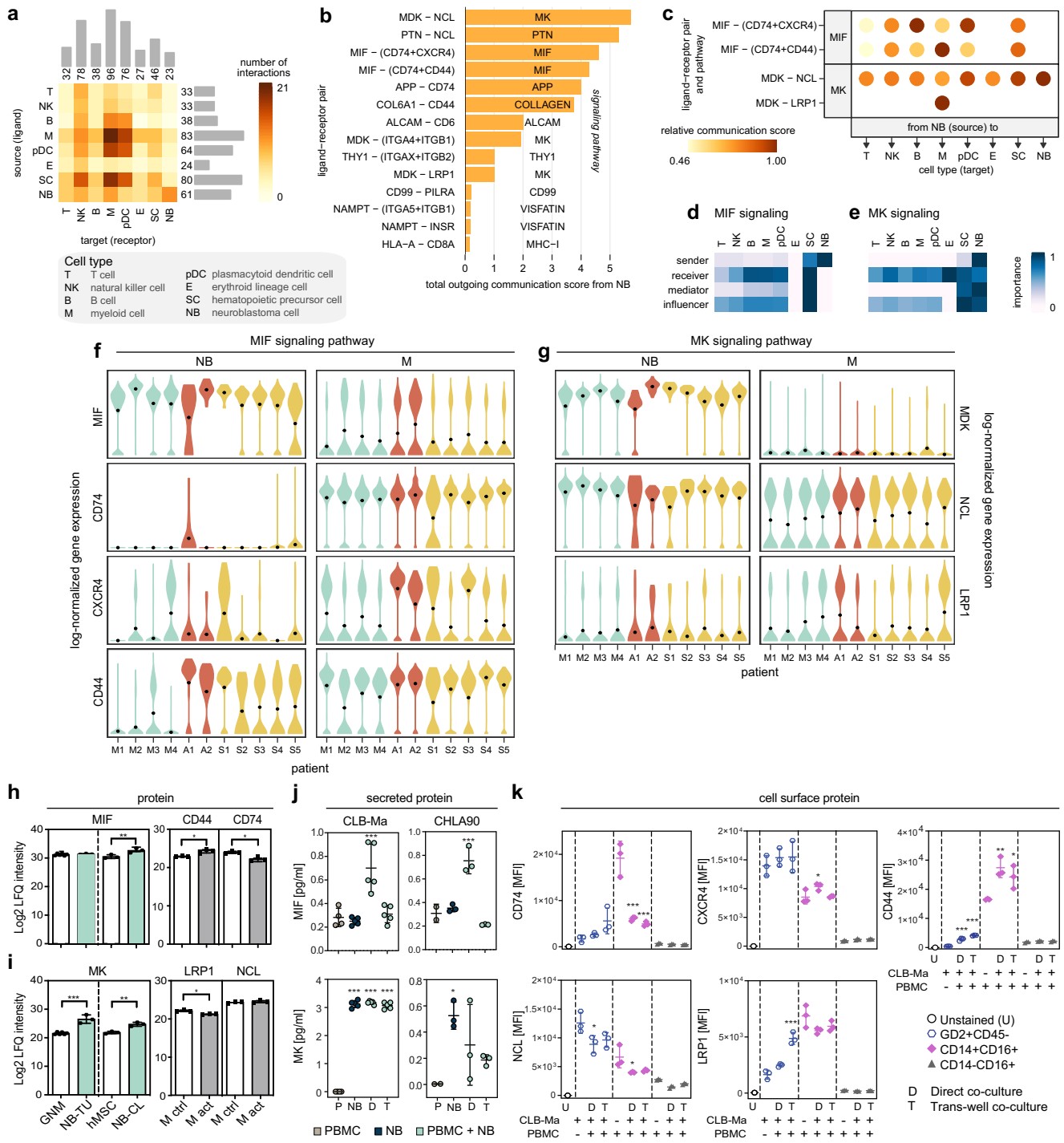

**h** protein — MIF, CD44, CD74
**i** MK, LRP1, NCL
**j** secreted protein — CLB-Ma, CHLA90
**k** cell surface protein

show that phenotypic plasticity is conserved upon metastasis and differs in MNA patients compared to *ATRX^mut* and sporadic NB subtypes, the latter two showing more pronounced transcriptional changes in the metastatic niche compared to the primary site.

## Interaction of neuroblastoma cells with the bone marrow microenvironment

To elucidate the crosstalk and communication pathways between NB and BM cell communities, we next inferred interactions between pairs of cells based on gene expression of annotated receptor-ligand pairs[50]. This revealed that NB cells express ligands, which are recognized by myeloid cells, and to a lesser extent by NK cells and plasmacytoid dendritic cells, suggesting a direct communication from NB cells to BM

cells, while communication from the microenvironment towards NB cells was marginal (Fig. 3a). Next, employing curated annotations for ligand-receptor pairs, we identified the principal communication pathways, which were midkine (MK), macrophage migration inhibitory factor (MIF), and pleiotrophin (PTN), and defined the molecules mediating interactions of NB cells to the microenvironment (Fig. 3b, c, Supplementary Data 3). In the MIF pathway, NB and stem cells are the main sources of MIF and predominantly communicate via this pathway with stem cells, myeloid, B-cells, and plasmacytoid dendritic cells, and to a lesser extent with NK and T-cells (Fig. 3d). Similarly, in the MK pathway, NB cells predominantly drive communication with erythroid, stem cells, and NK-cells, followed by plasmacytoid dendritic cells and myeloid cells, and to a lesser extent with B and T-cells (Fig. 3e).

**Fig. 3 | Cell-cell communication analysis of the NB tumors with the BM microenvironment. a** Number of ligand-receptor interactions of metastatic NB cells with the BM microenvironment. **b** Relative contribution of each ligand-receptor pair to the overall communication network. **c** Ligand-receptor pairs and their relative communication score in each cell type interaction. **d, e** Relative importance of each cell type based on the computed four network centrality measures of MIF and MK signaling networks, receptively. **f, g** RNA expression levels of ligands and receptors involved in the MIF and MK pathway, respectively, in individual patients. **h–i** Label-free protein quantitation of MIF and MK in ganglioneuroma (GNM, $n = 6$ biologically independent samples), primary NB tumors (NB-TU, $n = 3$ biologically independent samples), human mesenchymal stem cells (hMSC, $n = 3$ independent experiments), and NB cell lines ($n = 3$ biologically independent samples), and of CD44, CD74, LRP1, and NCL in primary monocytes stimulated with LPS ($n = 3$ independent experiments) and corresponding controls ($n = 3$ independent experiments). Data were subjected to two-tailed unpaired Student's $t$ test for MIF (GNM vs. NB-TU, $p = 0.46$ and hMSC vs. NB-CL, $p = 0.0025$) and

MK (GNM vs. NB-TU, $p = 0.0005$ and hMSC vs. NB-CL, $p = 0.0022$) or two-tailed paired Student's $t$ test (CD44, $p = 0.022$; CD74, $p = 0.027$; LRP1, $p = 0.017$; NCL, $p = 0.23$). **j** Secreted levels of MIF and MK as determined by ELISA in cell culture supernatants of CLB-Ma ($n = 4$ independent experiments for PBMCs, $n = 5$ independent experiments for all the other conditions) and CHLA90 ($n = 2$ independent experiments for PBMCs, $n = 3$ independent experiments for all the other conditions). **k** Mean fluorescence intensity (MFI) of cell surface proteins: CD44, CD74, CXCR4, LRP1, and intracellular receptor: NCL, as determined by flow cytometry in NB cells, co-cultured either directly with PBMCs or through a trans-well for 3 days. Cells were gated for GD2+ CD45−, CD14+ CD16+, and CD14− CD16+ populations ($n = 3$ independent experiments). Data (**j–k**) were subjected to one-way ANOVA and corrected using Dunnett's post hoc test for multiple comparisons. Asterisks indicate statically significant changes compared to control (PBMC or CLB-Ma): *$p < 0.05$, **$p < 0.01$, ***$p < 0.001$. Data are presented as mean ± standard error of the mean (**h–k**). Source data are provided as a Source Data file.

Furthermore, our data show that while tumor-derived MIF mostly acts in a paracrine manner on the BM microenvironment, MK also acts in an autocrine fashion on NB cells (Fig. 3d, e, Supplementary Fig. 3a, b). We observed a dominant crosstalk between NB and myeloid cells, where MIF signaling was mainly mediated by CXCR4, CD44, and CD74 (Fig. 3b, c, f), whereas MK signaling was primarily mediated through the MDK-NCL and MDK-LRP1 pairs (Fig. 3b, c, g). Both MIF and MK were also upregulated on protein level in NB tumors and cell lines compared to controls, and receptor expression was high on resting and activated monocytes (Fig. 3h, i). To experimentally test the interactions of NB with myeloid cells, we measured secreted levels of MIF and MK proteins in the supernatants of NB cell lines, peripheral blood mononuclear cells (PBMCs), and co-cultures of NB cells with PBMCs, either in direct contact or through a trans-well membrane. Levels of MIF were comparable between NB cell lines and PBMC controls, however, direct co-culturing resulted in elevated levels of MIF protein secretion, which was cell-cell contact-dependent (Fig. 3j). Levels of MK were higher in NB co-cultures compared to the PBMC cultures alone (Fig. 3j). Expression of cell surface receptors, CD44, CD74, CXCR4, LRP1, and NCL was determined by flow cytometry in CD14+ CD16+ and CD14− CD16+ myeloid cells, as well as GD2+ NB cells. All receptors were high in the CD14+ CD16+ PBMC population and were marked with significant differences between control group (PBMCs) and co-cultured settings (direct and trans-well) (Fig. 3k). The expression of these receptors was low or absent in the CD14− CD16+ population and did not show significant differences in co-cultures (Fig. 3k). Expression of CXCR4 and NCL was the highest in NB cells only (CD45− GD2+) compared to all other populations, which is in line with previous reports[51,52]. Moreover, the expression of CD44 and LRP1 was higher in co-cultured cells compared to control cells (Fig. 3k). Together, our data suggest that NB cells interact primarily with CD14+ CD16+ myeloid cells in the BM niche through paracrine MK and MIF, and these interactions are mediated by CD44, CD74, CXCR4 and LRP1, NCL receptors, respectively.

## Immune cell dynamics in patients with bone marrow metastatic neuroblastoma

To assess the impact of NB tumor infiltration on the BM microenvironment, we next investigated the composition of the BM microenvironment in MNA, $ATRX^{mut}$, and sporadic NB compared to controls. First, we assessed changes in cell type abundances, which revealed an enrichment in T- and NK cells, and a depletion of B- and myeloid cells in NB metastases compared to controls. Other cell types displayed marginal differences between NB patients and controls (Fig. 4a, Supplementary Fig. 4a). Second, gene expression analysis revealed an overall downregulation of gene expression in metastatic NB compared to controls, which was evident in all cell types, except myeloid cells (Supplementary Fig. 4b, Supplementary Data 4). Correlation analysis

showed that throughout the different BM microenvironment cell communities similar changes occur in $ATRX^{mut}$ and sporadic NB subtypes versus control, while microenvironment cells in the MNA subtype are distinct (Fig. 4b), a pattern reminiscent of the correlations observed in metastatic NB cells (Fig. 2c). Gene set enrichment analysis revealed enrichment for several inflammation-associated pathways, including TNFα and INFγ signaling, and upregulation of related genes e.g., $NFKB1$, $IL1B$, $SOD2$, $HLA-A$, and $HLA-DR$ in all subgroups (Fig. 4c, d). These changes occurred predominantly in B- and myeloid cells and represent typical M1-like features. In contrast, E2F and MYC targets, regulating cell cycle and proliferation, and their associated genes, e.g., $PCNA$, $MKI67$, and $CDK4$ were depleted/downregulated. Moreover, myeloid cells were characterized by hypoxia and epithelial to mesenchymal transition signatures (Fig. 4c, d, Supplementary Fig. 4c, Supplementary Data 5), indicative of an M2-like phenotype[53], which conveys early tumor progression, invasiveness, and resistance to chemotherapy[54]. Further investigation of myeloid subsets by subclustering into CD14+ classical and CD16+ non-classical monocytes, myeloid dendritic cells, and other myeloid cells revealed that the above-described pathways displayed similar trends across myeloid subtypes (Supplementary Fig. 5a–c, Supplementary Data 6). The differences between control and NB, as well as within NB subtypes, were most significant in CD14+ CD16− monocytes, marked by expression of typical M1 ($CXCL2$) and M2 ($CD163$, $TIMP1$, and $EREG$) markers, however, lacked expression of key macrophage markers (Supplementary Fig. 5b–f, Supplementary Data 7). Proteome data from primary monocytes and cell lines exhibiting M1 and M2 phenotypes, confirmed that BM metastasis-associated monocytes display M1 (elevated levels of NFKB1, IL1B, SOD2) and M2 (low levels of PCNA, MCM5) features (Fig. 4e, Supplementary Fig. 5f). Further characterization of patient BM metastases compared to control BM using multiplex imaging showed higher levels of CD14 (TLR4 binding LPS) and the β1 integrin CD29, mediating invasion in monocytes (Fig. 4f). Interestingly, we noted that protein levels of MHC class I and II members (HLA-A and HLA-DR) were lower when NB-cells were present (Fig. 4g). Cultivation experiments of NB cells with PBMCs demonstrated that in the CD45+ population, monocytes in direct co-cultures with NB cells were marked by high secretion of inflammatory M1 cytokines, INFγ, TNFα, IL-1β and M2 cytokines, IL-10 and TGFβ as well as a CD163+CD86dim phenotype, in line with M2-like cultures (stimulated with IL-4 and IL-10). This was in stark contrast to M1-like cultures (stimulated with IFNγ and LPS), which exhibited a predominantly CD163dimCD86high phenotype. Furthermore, monocytes co-cultured through a trans-well with NB cells displayed an increase in the fraction of CD14+ CD16+ population, however, again exhibiting an M2-like phenotype as marked by higher expression of CD163 (Fig. 4h–j). Finally, expression levels of MHC class I and II markers show that overall PBMCs co-cultured with NB cells display similar levels as under M2-like conditions (Fig. 4j). These data

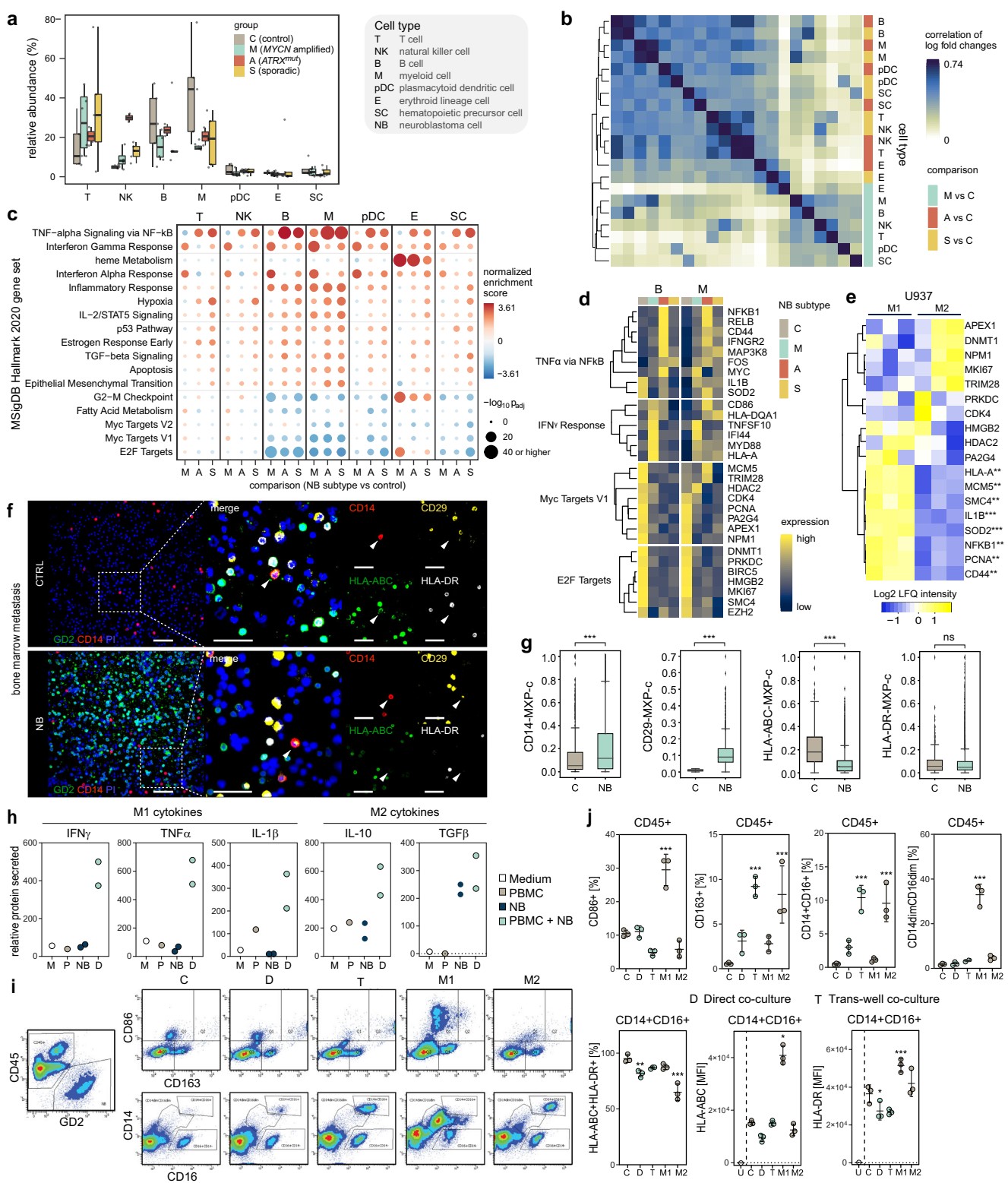

highlight a type of BM metastasis-associated monocyte that is induced by NB secreted factors and to a lesser extent by cell-cell contact, promoting tumor inflammation, mitogenic, and invasion signals.

### Chromatin landscape and heterogeneity in the neuroblastoma bone marrow metastatic niche

To investigate the underlying chromatin regulation associated with exhibited cell type-specific gene expression in BM metastases, we conducted scATAC-seq in the same matched patient sample set

($n = 16$). The assigned cell types in the scATAC-seq data (Fig. 5a) were concordant with the cell types identified in the scRNA-seq data (Fig. 1b and Supplementary Fig. 6a). To confirm the cell type assignment of scATAC-seq data, we investigated the accessibility of cell type-specific marker sets and lineage-specific transcription factor (TF) motifs in each cell type (Supplementary Fig. 6b). Integration of both scATAC- and scRNA-seq data for the genes associated with affected pathways in myeloid cells in NB patients compared to controls (Fig. 4c) revealed that loss of gene expression for MYC and E2F targets was associated

**Fig. 4 | Cellular composition of NB-infiltrated BM. a** Cell type abundance in NB subtypes (*n* = 4, 2 and 5 patients for *MYCN* amplified, *ATRX*^mut, and sporadic, respectively) compared to control (*n* = 5 patients). **b** Correlation analysis of gene expression changes between NB patients and controls for BM microenvironment cell types in NB subtypes. **c** Gene set enrichment analysis using MSigDB Hallmark 2020 database with genes sorted according to their log-fold change of expression relative to the control group. *p*-values are based on an adaptive multi-level split Monte-Carlo scheme as implemented in the R package fgsea. **d** Expression of exemplary genes in our patient cohort for the top two enriched/depleted pathways identified in **c. e** Label-free protein quantitation of targets identified in **d** in U937 cells (*n* = 3 independent experiments). **f, g** Protein expression in monocytes and macrophages derived from multiplex images of neuroblastoma BM samples with no (control, C, *n* = 3 patients) and high (>200 cells, *n* = 5 patients) NB cell infiltration. Scale bar, 45 μm. Wilcoxon–Mann–Whitney with FDR-corrected *p*-values: ns

not significant, \*\**p* ≤ 0.01, \*\*\**p* ≤ 0.001. **h** Secreted levels of M1 and M2 markers in co-cultured NB cells with PBMCs (*n* = 2 independent experiments). **i** Representative FACS plots of NB cells and CD45+ populations in co-cultured NB cells and PBMCs, and controls, along with PBMCs induced to acquire an M1 or M2 phenotype at day 3. **j** Percentage (%) of cells expressing the M1 marker, CD86 or the M2 marker, CD163 in CD45+, CD14+ CD16+, and CD14dim CD16dim populations as well as % and MFI of MHC class I and II markers (*n* = 3 independent experiments). Dots show the raw data (**a, g**) and boxes display the median value and 25 and 75% quartiles; the whiskers are extended to the most extreme value inside the 1.5-fold interquartile range. Data were subjected to two-tailed paired Student's *t* test (**e**) or one-way ANOVA (**j**, compared to control: PBMC or CLB-Ma), and corrected using Dunnett's post hoc test for multiple comparisons. Asterisks indicate statically significant changes, \**p* < 0.05, \*\**p* < 0.01, \*\*\**p* < 0.001. Data are presented as mean ± standard error of the mean (**j**). Source data are provided as a Source Data file.

with predominately closed chromatin. Interestingly, gain of expression of genes associated with TNFα and IFNγ pathways did not always equate with open chromatin (Fig. 5b, Supplementary Fig. 7a), which could be partly attributed to peak localization. Genome browser tracks of NFKB1 (TNFα pathway) and KDM2B (E2F target) highlight the differences in chromatin accessibility between NB patients and controls in the myeloid compartment (Fig. 5c). We inferred TF activity by motif enrichment analysis to further stratify how the organization of accessible chromatin in proximal and distal regions cooperatively acts in tandem with these chromatin-binding proteins. This revealed that NFκB-p65, which mediates the actions of TNFα stimulation was located in distal, but not proximal promoter regions (Fig. 5d), explaining, to some extent, the lack of correlation between scRNA-seq and scATAC-seq data in the TNFα pathway (Fig. 5b). Furthermore, we observed STAT3 activation, acting downstream of the M2 cytokine IL-10, where the latter was marked by an upregulation in BM metastases (Supplementary Fig. 7b, c). Similarly, co-culturing of NB cells with PBMCs resulted in higher levels of secreted IL-10 in both direct and trans-well conditions compared to NB cells or PBMCs alone (Supplementary Fig. 7d). High IL-10 potentially antagonizes the M1 cytokine IFNγ/STAT1 signaling[55] and explains the co-occurrence of M1 and M2 features in BM metastasis-associated monocytes. Moreover, myeloid cells, regardless of NB subtype, showed accessibility to AP-1 motifs and its dimers belonging to the Fos (Fos, Fosl2, Fra1/2) and Jun (JunB) families, in both promoter and distal regions, whereas NFκB and Bach1 motifs were only accessible in the distal regions. AP-1 is a core TF driving early myeloid lineage differentiation and deregulated expression has been reported in a host of malignancies[56]. Motifs of other key TFs involved in regulation of myelopoiesis and differentiation of myeloid cells, such as PU.1 and interferon response factors, e.g., IRF1[56] were marked by repressive chromatin, in both proximal and distal regions (Fig. 5d). TF footprinting analysis also confirmed enrichment of occupied regions by these TF in myeloid cells of NB metastases compared to control BM (Fig. 5e), suggesting reprogramming of bone marrow monocytes via key TF modules of myeloid lineage commitment and monocyte activation. Furthermore, network analysis links these key TF modules to open chromatin regions in genes associated with M2 polarization, tumor growth, and metastasis, including *IL-10*, *TIMP1*, and *EREG* (Supplementary Figs. 8–10). Thus, integration of scATAC-seq with scRNA-seq links epigenetically regulated myelo-monocytic lineage commitment and polarization with transcriptional changes resulting from external signals provided by and through tumor cells.

## Discussion

We present a single cell transcriptome and paired chromatin accessibility atlas of human BM metastases in NB. We find that while mutational status discriminates between healthy and cancerous cells in all NB subtypes, gene expression distinguishes primary tumors from metastasis in *ATRX*^mut and sporadic NB subtypes, whereas this difference is marginal in MNA tumors. Moreover, metastatic tumors were

associated with a distortion of the immune component in the BM, particularly of myeloid cells. NB cells signal to myeloid cells through MK and MIF pathways and promote an inflammatory environment, conveying a tumor-associated monocyte phenotype that is attributed to the rewiring of key signaling/transcription factor modules of myeloid lineage commitment and monocyte polarization (Fig. 6). Collectively, these data provide insights into the molecular and cellular architecture of NB across all subgroups and provide the basis for a therapeutic approach targeting tumor – myeloid cell interaction.

Investigations of the cellular origin in primary NB revealed that in low-risk NB tumor cells resemble normal sympathoblasts, whereas high-risk NB is marked by the presence of chromaffin-like cells and their progenitors (Schwann cell precursors), as well as the presence of cells with mesenchymal signatures[19–21,23]. Our analyses in adrenal medulla, primary NB tumors, and BM metastases, identified neuroblasts, cycling neuroblasts, and bridge cells as cellular correlates in MNA tumors, whereas non-MNA tumors were defined by the presence of late neuroblasts and chromaffin-like cells. This suggests that NB cells retain their phenotypic features upon metastasis, and that cell type composition is primarily determined by the NB subtype. Transcriptionally and epigenetically, NB cells are defined as undifferentiated mesenchymal cells and committed adrenergic cells[48,49]. Plasticity from an adrenergic to a more undifferentiated and resistant mesenchymal or NCC-like phenotype was observed under therapy in vitro, and in addition, the latter was enriched in pretreated and relapse primary tumors[48,49]. Our data show primarily a noradrenergic and adrenergic profile, and apart from chromaffin-like and SCP gene signatures, we did not detect a pronounced mesenchymal or NCC-like gene signature in metastatic NB cells, albeit these were present in other cell types. Recent studies reported the presence of cells with mesenchymal-like gene signatures in a minor subset of peripheral neuroblastic tumors and patient-derived xenograft mouse models[57,58]. Future studies will have to show whether these cellular transitions are dependent on cell intrinsic or other extrinsic factors and investigate their clinical relevance.

Myeloid cell infiltration has been reported in various tumors and is associated with poor clinical outcome[59], but reports in NB are discordant regarding their role in patient prognosis. Previous studies in NB showed that an increased infiltration of macrophages was inversely correlated with poor clinical outcome[30–34]. However, a recent study investigating the myeloid cell population heterogeneity in primary NB employing scRNA-seq demonstrated a positive correlation of various myeloid cell populations with pro-inflammatory cell states, and improved patient survival[37]. These recent advancements in our understanding of the tumor microenvironment in primary NB have paved the way for the development of anti-GD2 immunotherapy that is currently administered along with the conventional treatment in high-risk NB[60–62]. Here, we report a decreased presence of the myeloid cells at the metastatic niche, which was independent of NB subgroups. Interactions of tumor cells with the BM microenvironment involve

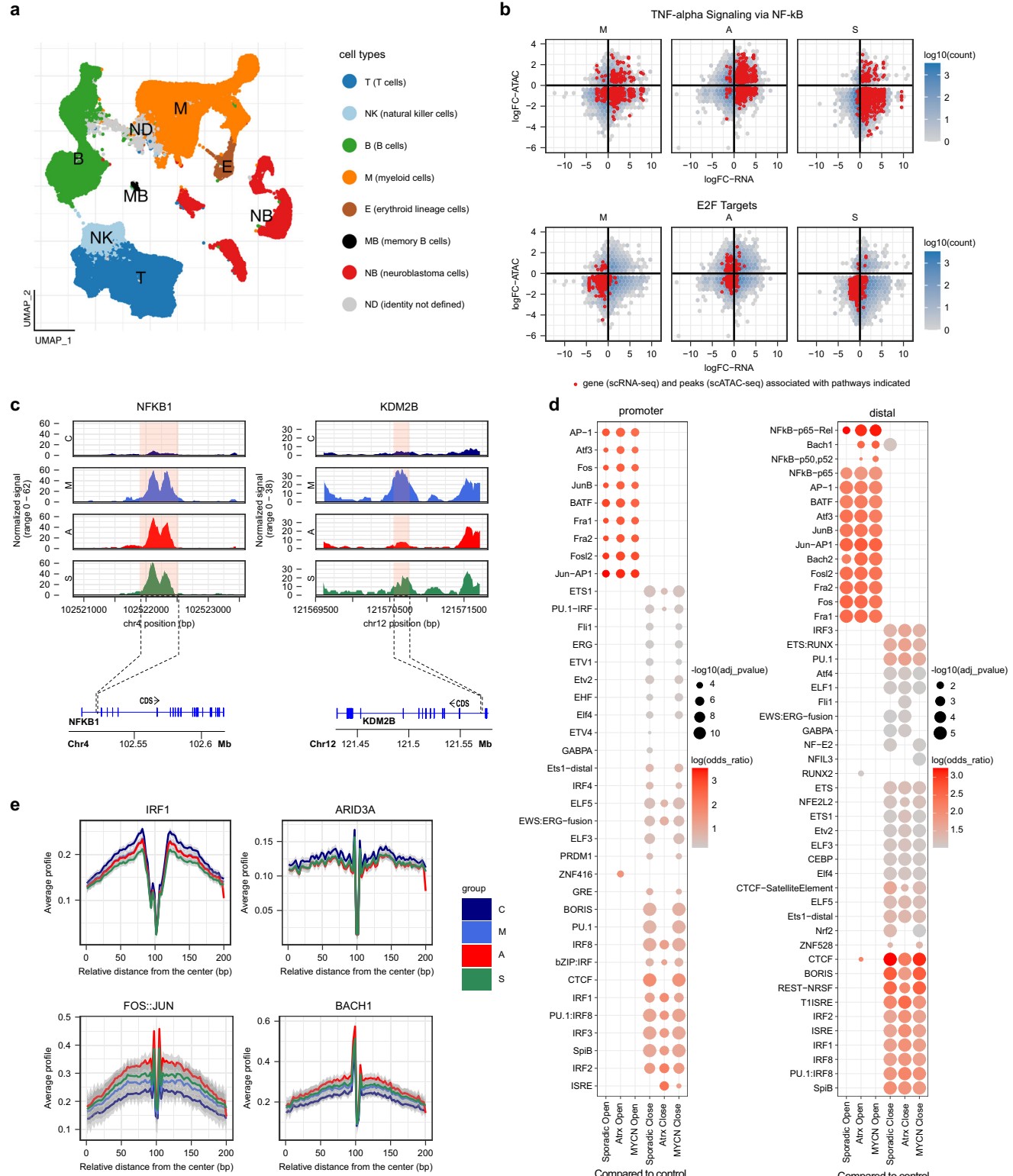

**Fig. 5 | Single cell regulatory landscape of NB-infiltrated BM. a** UMAP projection of scATAC-seq profiles of NB and BM cell types. Dots indicate individual cells, whereas colors indicate cell type identity. **b** Scatterplot of scRNA-seq and scATAC-seq log fold changes for genes in TNFα pathways and E2F targets. **c** Representative sequencing tracks for the NFKB1 and KDM2B loci show distinct pseudo-bulk ATAC-seq peaks in the NB subtypes compared to controls. The ATAC-seq data have been normalized with Signac[60] and the scale on the y-axis was chosen for optimal visualization of peaks for each sample. **d** Dot plot depicting odds ratio and adjusted

*p*-value (calculated with hyper-geometric statistical test implemented in HOMER) of motif enrichment in promoter and distal regions between NB subtypes (*n* = 4, 2 and 5 patients for *MYCN* amplified, *ATRX^mut^*, and sporadic, respectively) compared to control samples (*n* = 5 patients). **e** Differences in active binding of individual TFs between NB subtypes inferred with footprinting analysis, where the confidence interval represents variations between the patients of the group. Source data are provided as a Source Data file.

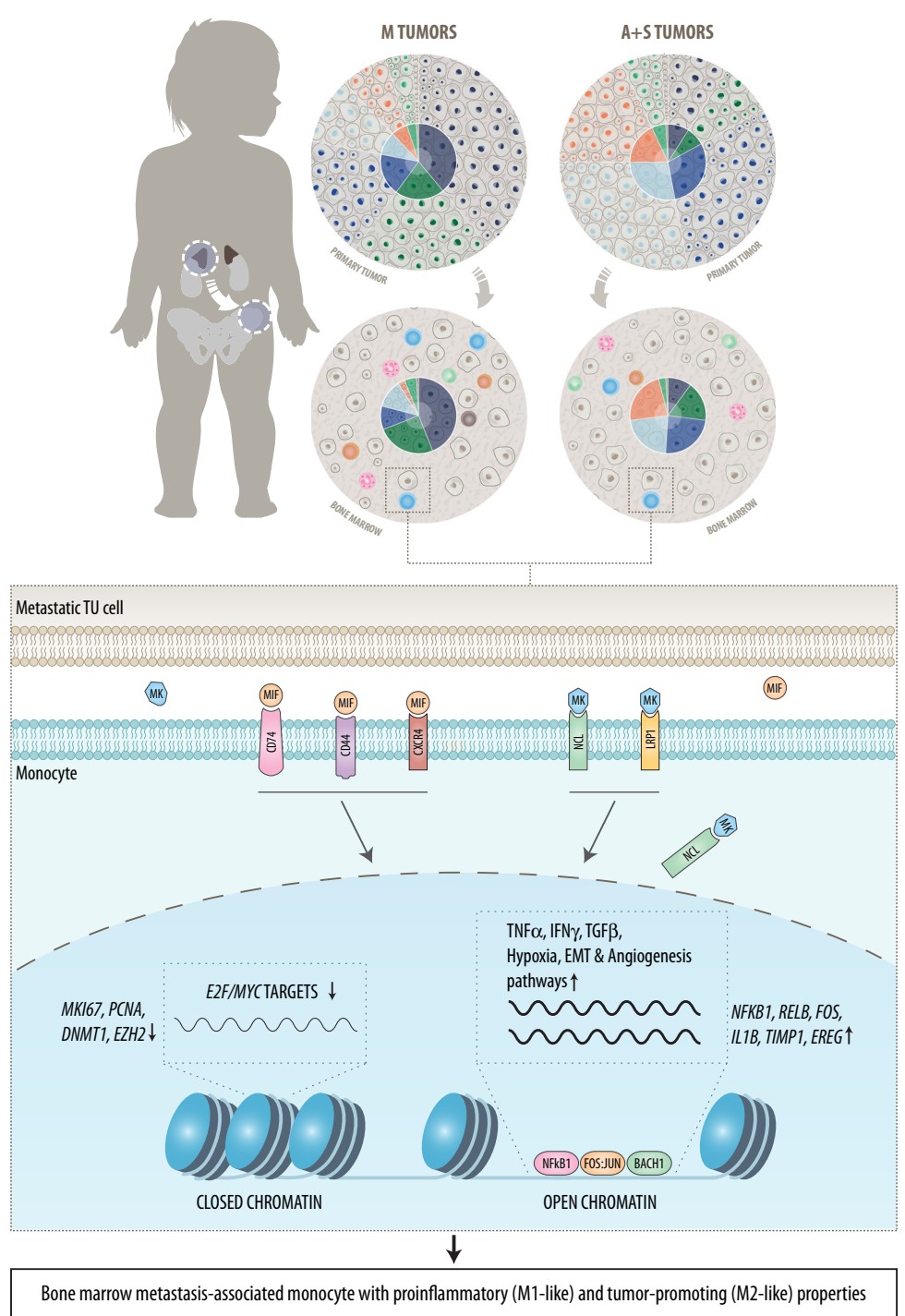

**Fig. 6 | Schematic illustration of the main findings and conclusions.** The figure depicts interactions between NB and myeloid cells in the bone marrow compartment, which are mediated through the MIF (Macrophage Migration Inhibitory Factor) and MK (Midkine) pathways. M, *MYCN* amplified; A, *ATRX^mut^*; S, sporadic.

preferred communication with myeloid cells through the MIF/CD44/ CD74/CXCR4 and MK/LRP1/NCL axes. Monocytes present M1 and M2 features indicated by aberrant pro- and anti-inflammatory core TF regulatory loops, pro-differentiation, and reduction of cell cycle genes as well as expression of tumor- and metastasis-promoting factors, such as TIMP1 and EREG[63,64] (Fig. 6). The role of MK has been described in physiological conditions, such as development and reproduction, as well as in pathological conditions like melanoma, where its over-expression leads to immunotherapy resistance by promoting

immunosuppressive myeloid cell differentiation as well as recruiting myeloid cells to the tumor site[65]. Similarly, MIF has been implicated in governing both inflammatory and tumor promoting functions in tumor-associated macrophages. In addition, MIF expression was higher in poorly differentiated NB, which led to increased MYCN expression in these tumors[66,67]. The receptors identified in this study have been under intense investigations in recent years as emerging therapeutics targets in various tumors. Indeed, small molecule inhibitors and antibodies against MIF and anti-CD74 antibody-drug-

conjugates are currently in phase I trials in autoimmune diseases and leukemia[68].

In conclusion, our work provides a cellular atlas of NB across all subgroups that defines the cellular states underlying each NB subgroup, disentangling determinants of intra- and inter-tumoral heterogeneity. Importantly, the ligand/receptors pairs identified in our study play pleiotropic roles in normal and disease settings and offer molecular targets for a therapeutic opportunity by disrupting tumor-to-microenvironment communication or monocyte polarization in the BM metastatic niche.

## Methods

### Human material
All patient material (Supplementary Table 1) used in this study was obtained from the CCRI Biobank after written informed consent was obtained for the use of left-over samples for research, including genetic analysis, from patients and/or their parents/guardians/legal representatives. Inclusion criteria: male and female patients aged 0 to 18 years with clinically, histologically, and biologically confirmed high-risk neuroblastoma with bone marrow metastasis or ganglioneuroblastoma or ganglioneuroma with no detectable bone marrow metastasis. Ethical approval for the CCRI Biobank and use of bone marrow aspirates and clinical data was obtained from the local institutional review board of the Medical University of Vienna (EK1216/2018, EK1853/2016). Peripheral blood of healthy donors was collected after obtaining informed written consent and approval of the local institutional review board of the Medical University of Vienna (EK 1150/2015). Patients were not compensated for study participation. Patients' sex was recorded, but not used to disaggregate the data, due to the limited number of patients and lack of statistical power. Gender was not assessed since NB is an early childhood tumor. Fresh bone marrow aspirates were collected in heparin or EDTA blood collection tubes at the time of diagnosis. Mononuclear cells were isolated by density gradient centrifugation as described previously[4] and cryopreserved in liquid nitrogen. Due to the limited sample size of biological materials collected from patients these materials are not available to be shared. The NB cell line, CLB-Ma was kindly provided by Dr. Valerie Combaret (Centre Leon Berard, France) and CHLA90 by the Children's Oncology Group Cell Line and Xenograft Repository and their identity was verified by SNParray or genome-wide sequencing analysis. The U937 cell line has been obtained from ATCC and authenticated using STR profiling. Cell lines routinely tested negative for the presence of mycoplasma, which was performed using a mycoplasma detection kit (MycoAlert, Lonza).

### Fluorescence-activated cell sorting (FACS) for scRNA-seq and scATAC-seq
Mononuclear cells of bone marrow aspirates were thawed and then resuspended in RPMI medium containing DNAse I (Roche), followed by a wash with cold 0.1% bovine serum albumin (BSA, Sigma) in phosphate buffered saline (PBS). Cells were then stained with the following antibodies (Supplementary Table 2): CD34, CD45, GD2 (kindly provided by Prof. Handgretinger, University of Tübingen), and L1CAM (CD171) in PBS/BSA 0.1% for 10 min at 4 °C. Finally, cells were stained with DAPI (Sigma) for 5 min at 4 °C and were passed through a 70 μm cell strainer into FACS tubes in PBS/BSA 0.1%. Acquisition and sorting of live cells were performed with FACSAria and the DIVA Software version 8.0 (Becton Dickinson). Sorted single cells were split into halves and were taken forward for scATAC-seq and scRNA-seq processing.

### Single-cell ATAC-sequencing nuclei isolation and library preparation
Nuclei isolation from bone marrow single cells was performed according to a 10x Genomics, Inc. protocol (CG000212). Briefly, cells in

PBS/BSA 0.1% buffer were centrifuged for 5 min at 4 °C, followed by incubation with 0.1x lysis buffer for 3 min on ice. Cells were then washed with cold 1x wash buffer and resuspended in diluted nuclei buffer to generate a nuclei stock concentration corresponding to 10,000 targeted nuclei recovery. Nuclei transposition and downstream library generation were performed using Chromium Single Cell ATAC Reagent Kits User Guide v1.1 (10x Genomics, Inc.) following their recommended protocol. Quality and concentration of libraries were determined by TapeStation (Agilent Technologies) and Qubit Fluorometric Quantification (Thermo Fisher Scientific). Samples were then normalized, pooled together, and sequenced (2x50bp) in NovaSeq (Illumina).

### Single-cell RNA-sequencing library preparation
Bone marrow single cells were taken forward for library generation using the Chromium Next GEM Single Cell 3′ Reagent Kits v3.1 (10x Genomics, Inc.) following their recommended protocol. In brief, 10,000 cells were used as input for the Gel Beads in-Emulsion (GEM) generation, where cDNA is generated and barcoded, followed by cDNA amplification and library construction. Quality and concentration of libraries were determined by TapeStation (Agilent Technologies) and Qubit Fluorometric Quantification (Thermo Fisher Scientific). Samples were then normalized, pooled together, and sequenced (2x50bp) in NovaSeq (Illumina).

### Copy number profiling by high-density single nucleotide polymorphism array (SNPa)
SNPa (Cytoscan HD, Thermo Fisher Scientific) data were generated during clinical molecular diagnostics from the corresponding primary tumor or, if available, the bone marrow sample that was processed for scRNA-seq. Metastatic tumor cells in the bone marrow were enriched by MACS sorting for GD2 as described previously[4]. DNA was extracted from purified bone marrow metastatic NB cells or fresh frozen tumor pieces and 50–200 ng DNA was used for SNPa analysis as described before[4]. Data analysis and CEL files were processed using ChAS 4.3 (Thermo Fisher Scientific). Copy number (log2 ratio) and bi-allele frequency tracks were visualized and predicted genomic segments (gains, losses, copy number) were manually curated using the IGV browser and plug-in VARAN-GIE[69]. TDF files were converted to BEDGRAPH format using IGV v2.9.4, and the logrr values in the latter files were directly plotted.

### Proteome data generation and analysis
We leveraged published data sets by[7,70] and re-analyzed the proteomics data of ganglioneuroma (GNM), primary NB tumors (NB-TU) and NB cell lines (NB-CL) as well as of human mesenchymal stem cells (hMSC). For the proteomic analysis of control and activated primary monocytes, peripheral blood mononuclear cells (PBMC) were isolated following Ficoll-Paque protocol from fresh blood of healthy donors. Next, washed PBMCs were subjected to magnetic-activated cell sorting using a positive selection kit (CD14 microbeads MACS, Miltenyi) to isolate the monocyte fraction following the recommended protocol. Monocyte population purity was determined to be >92% through FACS using a monoclonal antibody against human CD14 (clone 61D3, PerCP conjugated, Thermo Fisher Scientific). Monocytes were cultured in IMDM medium (GE Healthcare) supplemented with 10% fetal calf serum (GE Healthcare), 10ug/mL gentamycin (Sigma Aldrich), and 1.25ug/mL amphotericin B (Sigma Aldrich). To induce inflammatory response, monocytes were treated with 100 ng/mL LPS (Lipopolysaccharides from Escherichia coli 055:B5, γ-irradiated, Sigma-Aldrich) for 24 h. Isolation of cytoplasmic proteins, subsequent enzymatic protein digestion, as well as LC-MS analysis was performed as described before[71]. In brief, for the LC-MS/MS analysis of peptide samples derived from control and activated primary human monocytes, a Dionex UltiMate 3000 Nano LC system was used, which was coupled to

a Q Exactive Orbitrap mass spectrometer, equipped with a NanoSpray ion source (Thermo Fisher Scientific, Austria). Dried peptides were reconstituted in 5 μl 30% formic acid, containing the following synthetic peptides: Glu1-fibrinopeptide B, EGVNDNEEGFFSAR; M28, TTPAVLDSDGSYFLYSK; HK0, VLETKSLYVR; HK1, and VLETK(ε-AC)SLYVR were employed as quality controls. Samples were then further diluted with 40 μl mobile phase A. To preconcentrate the sample, peptides were loaded on a C18 2 cm × 100 μm precolumn following separation using a 50 cm × 75μm PepMap100 analytical column (Thermo Fisher Scientific). The flow rate was set at 300 nl/min and the injection volume was 10 μl. To achieve gradient elution of the peptides we increased the mobile phase B (79.9% acetonitrile, 20% H2O, 0.1% formic acid) from 8% to 40%, with a total chromatographic run time of 135 min, which included washing and equilibration. Mass spectrometric resolution on the MS1 level was set to 70,000 (at m/z = 200) with a scan range from 400 to 1400 m/z. The 12 most abundant peptide ions were selected for fragmentation at 30% normalized collision energy and analyzed in the Orbitrap at a resolution of 17,500 (at m/z = 200)[71].

To obtain M1- and M2-like macrophages, U937 cell line was cultured in RPMI medium (1X with L-Glutamine; Thermo Fischer Scientific) supplemented with 1% Penicillin/Streptomycin (Sigma-Aldrich) and 10% Fetal Calf Serum (FCS, Sigma-Aldrich). Induction of an M1-like phenotype was achieved by adding first 100 ng/mL Phorbol 12-myristate 13-acetate (PMA ≥ 99%, Sigma-Aldrich) for 48 h, followed by the addition of 100 ng/mL LPS (Lipopolysaccharides from Escherichia coli 055:B5, γ-irradiated, Sigma-Aldrich) for additional 48 h. In contrast, M2-like macrophage differentiation of U937 cells was achieved by first adding 100 ng/mL PMA for 24 h. Afterwards, 50ng/mL M-CSF (ImmunoTools) were added to the culture media for a total of 144 h. Cells were then incubated in fresh media containing 20 ng/mL IL-4 (ImmunoTools) for another 24 h. After differentiation, M1- and M2-like macrophages were washed twice with PBS and further incubated with 3 mL of serum free RPMI for 4 h. Thereafter, supernatants were precipitated using 12 mL cold EtOH (abs. 99%, −20 °C; AustroAlco) and cells were lysed in 200 μL of 4% SDC buffer containing 100 mM Tris-HCl (pH 8.5), immediately heat-treated at 95 °C for 5 min, and ultrasonicated. All samples were stored at −20 °C until further processing. For enzymatic protein digestion, an adapted version of the EasyPhos workflow was applied[72]. Briefly, 20 μg of protein was reduced and alkylated simultaneously using 100 mM TCEP and 400 mM 2-CAM, respectively, before a Trypsin/Lys-C mixture (1:100 Enzyme to Substrate ratio) was added for 18 h at 37 °C. For desalting, peptide solution was first dried to approximately 20 μL, mixed with loading buffer containing 1% TFA in isopropanol and loaded on SDB-RPS StageTips. After washing thoroughly, peptides were eluted with 60% ACN and 0.005% ammonium hydroxide solution, dried and stored at −20 °C until LC-MS analyses. LC-MS analysis and data processing was performed as described previously[73]. In brief, we used a Dionex UltiMate 3000 Nano LC system (Thermo Fisher Scientific) coupled to a timsTOF Pro mass spectrometer (Bruker), equipped with a captive spray ion source run at 1600V. Dried peptides containing four synthetic peptides [Glu1-fibrinopeptide B, EGVNDNEEGFFSAR; M28, TTPAVLDSDG-SYFLYSK; HK0, VLETKSLYVR; HK1, VLETK(ε-AC)SLYVR] were processed as described above. 5 μl of this peptide solution were concentrated on a pre-column (2 cm × 75 μm C18 Pepmap100, Thermo Fisher Scientific) at a flow rate of 10 μl/min using mobile phase A (99.9% H2O, 0.1% FA). The subsequent chromatographic separation was achieved on an analytical column (25 cm × 75 μm, 25 cm Aurora Series emitter column, IonOpticks) by applying a flow rate of 300 nL/min and using a gradient of 7% to 40% mobile phase B (79.9% ACN, 20% H2O, 0.1% FA) over 95 min, resulting in a total LC run time of 135 min, including washing and equilibration steps. The timsTOF Pro mass spectrometer was operated in the Parallel Accumulation-Serial Fragmentation (PASEF) mode. Trapped ion mobility separation was

achieved by applying a 1/k0 scan range from 0.60 to 1.60 V.s/cm², resulting in a ramp time of 166 ms. All experiments were performed with 10 PASEF MS/MS scans per cycle, leading to a total cycle time of 1.88 s. MS and MS/MS spectra were recorded using a scan range (m/z) from 100 to 1700[71].

## Multiplex immunofluorescence imaging and marker quantification

Differential expression analysis was performed on multiplex imaging-based single-cell data[35] of monocytes and macrophages derived from neuroblastoma bone marrow samples. Briefly, multi-epitope ligand cartography is based on repetitive cycles of antibody staining and photobleaching. After system start, four fields of view are selected and calibration (brightfield and darkframe) images are acquired. Prior to every staining and photo-bleaching cycle with the acquisition of the corresponding fluorescence tag and post-bleaching image, the slide is washed with PBS and a phase-contrast image is taken. Camera (ApogeeKX4, Apogee Instruments) and light source maintain the same position; the motor-controlled xy stage of the inverted fluorescence microscope (Leica DMIRE2, Leica Microsystems; x20 air lens; numerical aperture, 0.7) moves in between fields of view. Images with a resolution of 2018 × 2018 pixels are acquired, with one pixel corresponding to 0.45 μm at a 20× magnification. Protein expression, represented by the mean of the 20% highest pixel intensities was compared between cells derived from samples without and with tumor infiltration using the Wilcoxon–Mann–Whitney test with FDR correction. Data analysis and visualization was performed in python v3.9.12[74] using statannot v0.2.3 (https://github.com/webermarcolivier/statannot) and seaborn v0.11.2 packages[75], respectively.

## Co-culturing of NB cell lines with PBMCs

Here we employed two different NB cell lines: CLB-Ma (*MYCN* amplified) and CHLA90 (*ATRX*<sup>del</sup>). CLB-Ma cells were maintained under regular culturing conditions in RPMI medium (Thermo Fisher Scientific), supplemented with 1% Penicillin/Streptomycin (Thermo Fisher Scientific), 1.2% Sodium Pyruvate (PAN-Biotech), 2.8% HEPES Buffer (PAN-Biotech), and 10% FCS (Sigma-Aldrich). CHLA90 cell line was cultured in IMDM (Thermo Fisher Scientific) medium supplemented with 1% Penicillin/Streptomycin (Thermo Fisher Scientific), 1.2% Sodium Pyruvate (PAN-Biotech), 2.8% HEPES Buffer (PAN-Biotech), 10% FCS (Sigma-Aldrich), and 0.001% ITS (Insulin-Transferrin-Selenium, Thermo Fisher Scientific). PBMCs were isolated using a density gradient medium (Lymphoprep, StemCell Technologies) from fresh blood of healthy donors. Briefly, the blood samples were diluted with PBS (1:1) and then layered on top of the density gradient medium, followed by centrifugation at 800 × g for 25 min at 22 °C with break off. The middle layer containing the mononuclear cell fraction was carefully collected and washed twice with PBS at 300 × g for 10 min at 4 °C. After discarding the supernatant, cells were resuspended in RPMI or IMDM media, respectively, and were immediately taken forward for co-culturing experiments. NB cells were plated directly on the culturing dishes, followed by the addition of the PBMCs either in direct contact with NB cells or on trans-wells (6-well cell culture inserts, PET, 0.4 um, cellQart, Sterlitech) placed above NB cells, along with the respective controls, PBMCs and NB cells alone, and incubated for 72 h. To stimulate an M1 or M2 phenotype, PBMCs were incubated for 72 h either with INFγ (10³U/mL, PeproTech, cat#300-02) and LPS (100 ng/mL, Thermo Fisher Scientific, cat#00-4976-93) or IL-4 (20 ng/mL, Miltenyi, cat#130-093-922) and IL-10 (10 ng/mL, Miltenyi, cat#130-093-947), respectively.

## Receptor and MHC class I&II flow cytometry panels

Co-cultured cells as well as corresponding controls were harvested using Accutase (PAN-Biotech), washed once with cold PBS/BSA 0.1% (FACS buffer), resuspended in FACS buffer and Brilliant Stain Buffer

Plus (BD Biosciences), and then taken forward for FACS staining. For the receptor panel, cells were first incubated with the following extracellular antibodies: CD14, CD16, CD44, CD45, CD74, CD86, CD184 (CXCR4), CD163, GD2, and LRP1 (Supplementary Table 2) for 15 min at 4 °C. After a wash with FACS buffer, cells were fixed using Cytofix/Cytoperm (BD Biosciences) for 20 min at 4 °C, washed twice with 1x Perm/Wash (BD Biosciences), and then incubated with the intracellular antibody, NCL for 15 min at 4 °C. After two washes in 1x Perm/Wash, cells were resuspended in FACS buffer and analyzed immediately in FACS Symphony (BD Biosciences). For the MHC class I&II panel, cells were incubated with: CD14, CD16, CD45, CD171 (L1CAM), GD2, HLA-ABC, and HLA-DR (Supplementary Table 2) for 15 min at 4 °C, washed once with FACS buffer, and fixed using Cytofix/Cytoperm (BD Biosciences) for 20 min at 4 °C. Cells were washed twice with FACS buffer and immediately analyzed in FACS Symphony.

### Determination of cytokines and ligands by ELISA
The concentrations of IL-10 (cat#ab185986), MIF (cat#ab100594), and MK (cat#ab193761) were determined in undiluted cell culture supernatants of co-cultured cells and their controls by ELISA, all from Abcam. Samples were processed in duplicates following the manufacturer's instructions and measured in an analyte-dependent absorbance at 450 nm (reference wavelength: 570 nm), 2–5 min after addition of the stop solution with EnSpire Multimode Plate Reader (PerkinElmer). Standard curves were generated in parallel for each assay. Data were calculated by subtracting the signal from blank controls as well as the signal from the media control only.

### Determination of cytokines by multiplex proteome array
A multiplex proteome array (RayBiotech, cat#GS640) was used to determine the concentration of various secreted cytokines using undiluted cell culture supernatants of the co-cultured cells and corresponding controls according to the manufacturer's protocol. Briefly, 100 μl of sample was loaded into each well and incubated for 2 h at room temperature, followed by washes with buffers I and II. Samples where then incubated with a biotinylated antibody cocktail for 2 h at room temperature, washed with buffers I&II, and incubated with a Cy3 equivalent dye-streptavidin conjugated overnight at 4 °C. Samples were washed with buffers I&II, air dried, and acquired using GenePix 4000B microarray scanner (VWR). Data were calculated by normalizing the signal for each protein to the average positive control signal.

### Single-cell (sc)RNA-seq analysis
Unless otherwise stated, all analyses were conducted in R (v4.0.5)[76]. Figures were plotted with ggplot2 (v3.3.3)[77] and ComplexHeatmap (v2.6.2)[78].

**Preprocessing and quality control of scRNA-seq data.** Transcript counts were obtained by processing FASTQ files with Cell Ranger (v3.0.2, 10x Genomics, Inc.) using GRCh38 [https://www.ncbi.nlm.nih.gov/assembly/GCF_000001405.39] as the reference genome, which yielded an initial dataset comprising of 106,864 cells. Detailed information on sequencing results is provided in Supplementary Data 1. Only cells matching the following criteria (as calculated by Seurat v4.0.0[79]) were included for downstream analyses: >200 and <500 features, as well as <10% reads mapped to mitochondrial genes. Moreover, doublets with a binary classification-based doublet score >0.8 (as calculated by scds v1.6.0[80]) were discarded. Eventually, quality control filtering yielded a final dataset consisting of 80,789 cells. Cell-free mRNA contamination was removed via SoupX (v1.5.0)[81].

**Dimensional reduction and clustering of scRNA-seq data.** Raw counts were log- and size factor-normalized and scaled, followed by dimensional reduction via principal component analysis, as implemented in monocle3 (v0.2.3.0)[82]. Principal components were subsequently used as input for batch correction by matching mutual nearest neighbors, employing the function 'reducedMNN' in batchelor (v1.6.2)[83]. The Uniform Manifold Approximation and Projection (UMAP) method for dimensional reduction (uwot, v0.1.10)[84] was applied to the resulting batch-corrected principal component scores. Finally, Leiden clustering[85] was performed on UMAP coordinates. These steps were conducted via the monocle3 functions 'align_cds,' 'reduce_dimension,' and 'cluster_cells.'

**Cell type classification of scRNA-seq data.** Cells were classified via SingleR (v1.4.1)[86], using fine-grained labels in the following five reference datasets provided by celldex (v1.0.0): Human primary cell atlas[87], Blueprint/ENCODE[88,89], Database of Immune Cell Expression[90], Novershtern hematopoietic data[91], and Monaco immune data[92]. Following this initial classification of individual cells, all cells in a cluster were annotated with a single cell type, which was determined by a majority vote on cell ontology identifiers associated with the cell type labels in each cluster (Fig. 1c, Supplementary Fig. 5a).

**Copy number variation calling from scRNA-seq data.** Copy number variations (CNVs) were inferred from scRNA-seq data via infercnv (v1.6.0)[93]. To this end, non-NB cells were labeled as normal reference cells. The baseline expression in normal cells was then subtracted from both the tumor cells as well as the normal cells, yielding residual expression values. For this calculation, a dynamic threshold for noise filtering was applied (arguments 'denoise = TRUE' and 'sd_amplifier = 1.5' for infercnv's 'run' function; Supplementary Fig. 1d, e). Subsequently, copy number alteration regions were predicted via a six-state Hidden Markov Model (arguments 'HMM = TRUE' and 'HMM_type = "i6"'). For each region, its posterior probability of alteration status was determined by a Bayesian network latent mixture model (argument 'BayesMaxPNormal = 0.5'; Fig. 1e, Supplementary Data 2). Regions whose posterior probability of being in a normal state exceeded 0.5 were discarded.

**Assignment of gene signature scores in scRNA-seq data.** Gene signature scores (Supplementary Fig. 2a, b) were calculated from raw counts based on the Mann–Whitney U statistic, as implemented in Ucell (v1.0.0)[94]. Adrenergic and mesenchymal signatures were obtained from Supplementary Table 2 in van Groningen et al.[49], while noradrenergic and neural crest cell-like signatures were taken from Fig. 1h in Boeva et al.[48].

**Comparison to adrenal medullary cells in scRNA-seq data.** Similarity of NB cells to adrenal medulla cells (Fig. 2b, Supplementary Fig. 2d) was determined via SingleR (v1.4.1), using adrenal medulla data downloaded from https://adrenal.kitz-heidelberg.de/developmental_programs_NB_viz/[21] as reference dataset. Differentially expressed genes between pairs of labels were detected by the Wilcoxon ranked sum test (argument 'de.method = "wilcox"').

**Correlation of pseudobulk scRNA-seq data.** Gene expression profiles of primary and metastatic tumor cells (Fig. 2c and Supplementary Fig. 2e) were compared by (1) selecting NB cells in all samples that were annotated as high-risk in the dataset by Dong et al.[20] and all NB cells in our dataset; (2) performing scaling normalization within each dataset employing the function 'multiBatchNorm' in batchelor (v1.6.2)[83], and combining the datasets; (3) modeling per-gene variance and selecting all highly variable genes using the functions 'modelGeneVar' and 'getTopHVGs' in scran (v1.18.5); (4) aggregating raw counts at the sample level via the function 'aggregateData' in muscat (v1.4.0)[95] to obtain pseudobulk data; and (5) calculating Pearson correlation coefficients $r_{ij}$ (R function 'cor') and using these coefficients as distances $1 - r_{ij}$ for hierarchical clustering (R function 'hclust(method = "complete")').

**Analysis of cell-cell interactions in scRNA-seq data.** Cell-cell interactions (Fig. 3a–e, Supplementary Fig. 3a, b) were inferred from scRNA-seq data via CellChat (v1.1.0)[50], obtaining human ligand-receptor interactions from the CellChatDB database.

**Cell type abundances in scRNA-seq data.** Changes of cell type abundances in samples from NB patients compared to control samples (Supplementary Fig. 4a) were determined by testing whether the fractional abundance of a cell type in a patient was greater than the fraction of this cell type in the control samples. Fisher's exact test (R function 'fisher.test') was used to obtain odds ratios and p-values. Multiple test correction was performed using the Benjamini–Hochberg method (R function 'p.adjust').

**Differential gene expression and gene set enrichment analysis in scRNA-seq data.** Genes that were differentially expressed between the control group and each of the NB groups or between MNA and non-MNA tumors (Supplementary Data 4 and 7) were determined by fitting gene-wise negative binomial mixed models using large-sample approximation via the nebula package (v1.1.7)[96] in R. Within nebula, patients were modeled as random effects, and the tumor infiltration was added as a fixed effect to correct for differences in tumor infiltration. Log-fold changes of genes that were expressed in at least 5% of cells of the compared groups were subjected to gene set enrichment analysis (Fig. 4c, Supplementary Fig. 5c, Supplementary Fig. 7b, and Supplementary Data 5 and 6) as implemented in fgsea (v1.16.0)[97] in R, which estimates p-values using an adaptive multi-level split Monte-Carlo scheme and calculated corrected p-values using the Benjamini–Hochberg method. Gene sets comprised MSigDB_Hallmark_2020 and TRRUST_Transcription_Factors_2019, which were downloaded from Enrichr[98]. Genes were ranked according to their log-fold change of expression relative to the reference group. Moreover, log-fold changes in each cell type and sample were compared by calculating Pearson correlation coefficients $r_{ij}$ (R function 'cor') and using these coefficients as distances $1 - r_{ij}$ for hierarchical clustering (R function 'hclust(method = "complete")' as reported in Supplementary Fig. 4b.

## Single-cell (sc)ATAC-seq analysis

**Preprocessing and quality control of scATAC-seq data.** Initial processing of individual samples was performed using Cell Ranger ATAC 1.2.0 pipeline (10x Genomics, Inc.) and aggregated with Cell Ranger ATAC Aggregator. Detection of open chromatin region (OCR) was conducted with Cell Ranger ATAC. The initial preprocessing and filtering of scATAC-seq data was done with Signac[99]. According to standard recommendations, cells with 500–25000 fragments in peak regions and at least 25% of reads in peak region, nucleosome signal score less than 3, and TSS enrichment score greater than 1 were retained for further analysis (Supplementary Fig. 11a–e). This quality control filtering yielded 71,857 cells, which were used for subsequent analysis (Supplementary Data 8).

**Normalization, imputation, and dimensionality reduction of scATAC-seq data.** Normalization of the data was done with the Signac method of term frequency - inverse document frequency (TF-IDF) method with default settings. To account for the sparsity of the dataset and inherent signal loss in scATACseq experiments, we performed a non-negative matrix factorization (NMF) using coordinate descent algorithm with scOpen that imputes accessibility scores. Next, dimensionality reduction was performed using TF-IDF+NMF implemented in scOpen[100], and clustering was performed using Signac's FindNeighbors and FindClusters functions.

**Cell type classification of scATAC-seq data.** To classify cell types in scATAC-seq data, a gene-activity matrix was inferred combining open chromatin region 2 kb upstream of the TSS[99]. The R package Ucell[94] was used to calculate gene signatures for each cell using the gene activity matrix and cell markers from PanglaoDB[47], and a study describing the bone marrow niche Baryawno et al. 2019[101].

**Analyses of differential open chromatin regions between cell types in scATAC-seq.** The differential open chromatin regions for each cell type were identified with Signac function FindMarkers. The default parameter of min.pct (minimum percent of cells to consider) was changed to 0.05 from 0.1, considering the sparse nature of scATACseq data. The LR (logistic regression) test was used to test for the significance and perform multiple test corrections. Only regions with adjusted p-value ≤ 0.05 were further considered for the motif analysis as per Signac vignette titled 'Motif analysis with Signac' (https://satijalab.org/signac/articles/motif_vignette.html)[99] with combined motif frequency matrix included in JASPAR2020[102].

**Analyses of differential open chromatin regions between NB patient groups in scATAC-seq.** The differentially accessible regions between the microenvironment in NB patient groups and controls were identified by fitting region-wise negative binomial mixed model using nebula (v1.1.7)[96] in R, followed by multiple test correction. The resulting differentially accessible regions were filtered for adjusted p-value ≤ 0.05 and log2FC = 1. Annotations to the differential open chromatin regions were added with R packages TxDb.Hsa-piens.UCSC.hg38.knownGene, org.Hs.eg.db, ChIPseeker[103]. The open and close chromatin regions from each comparison were then separately subjected to motif analysis with findMotifsGenome.pl script from the HOMER package (version v3.0)[104], which includes multiple test correction. Only promoter regions were considered for motif analysis (promoter region is defined as regions ±300 bp from the TSS). From the motifs identified by the HOMER script only motifs with an adjusted p-value < 0.05 and log_odds_ratio >05 were considered significant and are shown in Fig. 5d.

**Transcription factor footprinting.** To identify the TFs bound to DNA at the time of enzymatic cleavage we adapted footprinting analysis with HINT-ATAC[105]. For each NB subtype, we combined reads from the single cells for each cell-type cluster separately and created pseudo-bulk ATAC-seq libraries. All footprint-supported motifs from the JASPAR database were used for footprint analysis.

**Integration of scRNA-seq and scATAC-seq.** We applied GLUE algorithm (Graph Linked Unified Embeddings), which uses peak and gene proximity correlation within linear genome as prior knowledge to integrate our scATAC-seq and scRNA-seq data (Similar to priors in Bayesian analysis to get regulatory inference)[106]. We imported scRNA-seq and scATAC-seq data in the anndata format and filtered the dataset to features found in at least 3 cells. GLUE was run on scATAC-seq data with 100 components and 15 iterations. To map peaks to genes, we only used promoter peaks as annotated by Gencode gencode.v40.chr_patch_hapl_scaff.annotation.gtf.gz. Once the links between peaks and genes were established, the embeddings from scRNA-seq were assigned to scATAC-seq to deal with the sparseness of the scATAC-seq data. The cosine similarity between the embeddings was used to evaluate the peak-gene association strength. Neighbors of scRNA-seq and scATAC-seq data were calculated using the cosine similarity between embeddings. The p-values for the peak-gene associations were obtained by comparing with a null distribution of shuffled embeddings. The UMAPs of data integration are shown in Supplementary Fig. 11f, g. Employing GLUE embeddings we then inferred the gene regulatory network within myeloid cells (Supplementary Figs. 8–10).

## Reporting summary

Further information on research design is available in the Nature Portfolio Reporting Summary linked to this article.

## Data availability

Raw scRNA-seq and scATAC-seq data have been deposited in EGA and are accessible through accession number EGAS00001006106. The data are available under restricted access due to data privacy laws and access can be obtained by contacting the Data Access Committee. Data deposited at EGA will be made available to researchers for non-commercial use only upon request and without a time limit. Count matrices have been deposited in NCBI's Gene Expression Omnibus and are accessible through GEO SuperSeries GSE216176. The MELC multiplex imaging and single-cell data of our neuroblastoma cohort is available at https://doi.org/10.5281/zenodo.6621045[107]. SNPa data and R objects are available from https://doi.org/10.5281/zenodo.7707614[108]. The mass spectrometry proteomics data have been deposited to the ProteomeXchange Consortium (http://www.proteomexchange.org/) via the PRIDE partner repository (PXD036979 and PXD036972). Source data are provided with this paper.

## Code availability

Code for performing the analyses and generating all figures is available from GitHub (https://github.com/csbg/neuroblastoma) and has been archived to Zenodo (https://doi.org/10.5281/zenodo.7867892)[109].

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

## Acknowledgements

This work was supported by Vienna Science and Technology Fund (WWTF#10.47379/LS18111) to S.T.M., N.F., and R.L., by Austrian Science Fund (FWF#P35841-B, FWF#I4162) to S.T.M., by Austrian Science Fund (FWF#P35072) to I.S.F, by Vienna Science and Technology Fund (WWTF#LS18-049) to E.M.T. and C.B., and the St. Anna Kinderkrebsforschung e.V. We thank all the patients and donors who participated in this study. We thank the St. Anna Children's Cancer Research Institute Flow Cytometry Core Facility for their assistance in cell sorting. We thank Markus Wiederstein and Peter Lackner at Paris Lodron University Salzburg for providing high-performance computing infrastructure, and the Biomedical Sequencing Facility at CeMM for sequencing and initial bioinformatic processing. We are grateful to Drs. Eva König, Florian Halbritter, David Kreil, Christian Argueta, and Kimberlie Rabidou for fruitful scientific discussions.

## Author contributions

S.T.M. and N.F. conceived, designed, and supervised the execution of the entire study. I.S.F. conceived, designed, and performed most of the experimental work, analyzed, and interpreted the data. W.E.S., R.D., and S.S. were responsible for the bioinformatics data analyses and respective data visualizations. S.T.M., D.T., M.T., E.B., M.F., M.R., and L. Shaw performed experimental work. D.L. performed the multiplex imaging and associated data analyses. A.B., L. Skos, and C.G. designed and performed proteome analysis. F.R. assisted in data visualization as well as illustrations. R.L., S.T.M., F.R., and M.B. contributed essential samples and clinical data. E.M.T., C.B., W.W., and M.F. supported the experimental design. I.S.F., S.T.M., and N.F. co-wrote the manuscript with input from W.E.S., R.D., and M.F. All authors have revised the manuscript and have read and agreed to the published version of the manuscript.

## Competing interests

All authors declare no competing interests.
