## [Peer Review File · Nature Communications]

REVIEWER COMMENTS

Reviewer #1 (Remarks to the Author): expertise in single cell sequencing

Fetahu and colleagues have generated a substantial amount of information at single cell level of human BM metastases in NB. They have performed a thorough analysis determining cell type landscape, NB subtype composition, cell to cell communication between cell types and further analyzed the different myeloid sets. Besides, they correlated this transcriptome information with epigenetics.

The data is really valuable and the analysis performed well detailed and justified. I'm only concerned about the following points:

Integration: authors have made use of batchelor (FastMNN) to perform integration of the samples. Tran et al, 2020 [1] benchmarked these kinds of integration methods, showing better performance from others (LIGER, Harmony,...) than FastMNN. The authors do not justify why they used this method. Could the authors discuss the effect of the analysis on integrating with different methods?

There is a need for more references. E.g: the authors claimed that "[...] we identified a cluster of cells classified as neurons, expressing key NB markers [...]". Where did the authors obtain these markers?

FACS Ext fig 1B. How did the authors obtain the NB cells in the FACS? GD2-FITC MARKER? Where are the source data numbers for this fig?

Authors claim that "[...] CNV identified from bulk tumor profiling by SNP-array further corroborated our tumor cell assignment". Fig1e does not show clearly this. There are several regions in the scRNA-seq that are not reflected in the SNP array, this latter technique is supposed to be more sensitive than inferring CNV from scRNA-seq

Extended Fig1c,d: Did the authors normalize data from Fig1e with the same cells? This is what it looks like from the description. If this is the case is normal that no variations were detected. Authors should take other healthy bone marrow cells to call for variations.

The authors show greater expression of noradrenergic markers than the rest. Can the authors test this enrichment?GSEA?

Authors claim: "The differences between control and NB subtypes [...] were most significant in CD14+CD16- monocytes [...]". The authors show UMAP expression of some of these genes between control and NB subtypes. It would be necessary to quantify and test these differences.

Regarding characterization of the myeloid component within the BM compartment, there are an important amount o myeloid cells non-characterized using the reference-based approach. Could the authors elaborate on the upregulated genes in this cluster?

Authors claim: "[...] loss of gene expression for MYC and E2F targets was associated with predominately closed chromatin". According to Fig5B, these could be for M and S tumors, but not for A. Could the authors correlate these associations and test them? Additionally, switch the figures in Fig5B to place them in accordance with the text

Authors have focused their attention on the myeloid cells, but they also found out that B cells are enriched in inflammatory pathways and depleted. Why the authors have not further investigated changes in the lymphoid compartment, both B and T cells? This seems to have an important impact on NB biology [2]

Github link was not accessible to me

[1] Tran, H.T.N., Ang, K.S., Chevrier, M. et al. A benchmark of batch-effect correction methods for

single-cell RNA sequencing data. *Genome Biol* 21, 12 (2020). <https://doi.org/10.1186/s13059-019-1850-9>

[2] Verhoeven BM, Mei S, Olsen TK, Gustafsson K, Valind A, Lindström A, Gisselsson D, Fard SS, Hagerling C, Kharchenko PV, Kogner P, Johnsen JI, Baryawno N. The immune cell atlas of human neuroblastoma. *Cell Rep Med*. 2022 Jun 21;3(6):100657. doi: 10.1016/j.xcrm.2022.100657. Epub 2022 Jun 9. PMID: 35688160; PMCID: PMC9245004.

Reviewer #2 (Remarks to the Author): expert in brain cancer bone metastasis

The Letter to Nature communications entitled "Dissecting the cellular architecture of neuroblastoma bone marrow metastasis using single-cell transcriptomics and epigenomics unravels the role of monocytes at the metastatic niche" by Fetahu et al. presents a very interesting and informative investigation into the metastatic niche of high risk neuroblastoma using scRNA and scATAC seq approaches. They utilize patient material from 11 different diagnostic cases that represent MYCN amplified and non-amplified versions of neuroblastoma and did an excellent job presenting the rationale for their approach to analyze these samples.

Overall, this is a very well executed work and I would recommend publishing without major revisions. The conclusions of the manuscript are consistent with current concepts of neuroblastoma biology and shed particular insight into the monocyte/macrophage components of bone marrow interacting with metastatic cancer cells. These results are noteworthy as pharmaceutical and targeted approaches to interfering with tumor suppressive immune cells are rapidly being developed and are expected to help improve the outcome for difficult to treat metastatic solid tumors such as Neuroblastoma.

Two points:

1) The MYCN amplified tumors have a much higher percentage of tumor involvement in the bone marrow compared to the atrx and sporadic subtypes. – How this may confound the sequencing data analysis should be directly addressed (Facs %37-56 versus 2-25% from table 1). It is not clear to me how this increased in tumor cell representation was accounted for in the data analysis.

2) The authors highlight the MK and MIF pathways based on bioinformatic assessment of ligand/receptor expression across the various cell types (figure 3). To this reviewer the inferred interactions are interesting but the validation of this type of gene expression at the protein level and individual receptor ligand pair level is lacking. Since we know that many receptors or pseudoreceptors are expressed that do not signal at all in various tumor subpopulations, I find the data in figure 3 weaker than the rest of the manuscript. However, it remains to be seen how useful this type of bioinformatic inference is.

3) While the published data on core regulatory networks and TF regulation in Neuroblastoma is sighted, it would be helpful if the discussion expanded on how the data here reflects the previous work (which was mostly in cell lines, and I believe did not include metastatic samples).

One comment – while MYCN amplified and ATRX mutant tumors appear to have defined drivers, it seems premature to term all the rest of Neuroblastoma as Sporadic if it lacks a MYCN or ATRX mutations..

- Are there any flaws in the data analysis, interpretation and conclusions? Do these prohibit publication or require revision? I do not find any obvious issues with the data in this manuscript
- Is the methodology sound? although I am not expert in the integration of ATAC and RNA seq sc data sets, the presentation is very clear and methodology is sound.
- Does the work meet the expected standards in your field? Yes
- Is there enough detail provided in the methods for the work to be reproduced? Yes

Overall, this work sheds additional light on tumor/immune cell interactions utilizing well defined patient samples that are representative of the typical spectrum of Neuroblastoma tumors. In addition, they highlight novel and potentially targetable immune subpopulations (pending validation).

Reviewer #3 (Remarks to the Author): expert in scRNA-seq and ATAC-seq bioinformatics techniques

In this study, Irfete et al. performed scRNA-seq and scATAC-seq of cells in bone marrow from 5 samples of benign tumors without BM metastases and 11 samples across metastatic subtypes. Firstly, the data of the first single cell transcriptome and paired chromatin accessibility atlas of human BM metastases from NB are quite precious. Secondly, the downstream analysis covers many interesting layers, including CNV analysis, cell-cell communication, and so on. Thirdly, some experiments, such as SNP array and protein quantitation, in addition to sequencing data, are included for validation. Overall, Irfete et al. did a valuable study, but the following points could still be improved.

Major comments:

1. The introduction to the background is a bit confusing and vague, leaving no idea what biological problem the author is trying to solve. An interesting question is what determines or influences the BM metastases from NB? What patients will and will not metastasize? Are there public sequencing data of NB primary tumors with and without metastasis available for analysis? Is metastasis determined by the primary site (the cancer cell is too strong) or by the metastasis site (the normal cells in metastasis site are too weak)?
2. The authors only showed the basic quality description from cellranger in extended data table, and more quality control analyses of scATAC-seq are needed. For example, the fragment size distribution and TSS enrichment score are recommended.
3. Since the authors have identified some interesting molecules or pathways in BM metastases from NB, such as VEGF and EREG, the regulatory network can be further constructed from scRNA-seq and scATAC-seq. In addition, as the enriched motif are listed in Fig5d, the author could find the target genes of these TFs by inferring the regulatory network from scRNA-seq and checking the peak-to-gene linkages from scATAC-seq, which may link the two types of sequencing and provide insights in BM metastasis.
4. The authors are supposed to integrate scATAC-seq and scRNA-seq, which may lead to interesting new clues. There are many tools for integrating multi-omic sequencing. For instance, the authors could annotate clusters of scATAC-seq using scRNA-seq and identify predicted cis-regulatory elements. This may also explain to some extent which important changes are caused by epigenetics and which are caused by other influences such as CNV.
5. There is some ambiguity in the description of whether all cells or certain types of cells were used in a particular analysis, such as CNV inferred from scRNA-seq in Fig1e and ATAC-seq signal in Fig5c. And why not view them separately by cell type?
6. Many of the pictures are not clear enough, such as Fig1a and schematic illustration in other figures. And some colors in figure are too similar to distinguish, such as colors representing sample C3, M3 and S3 in extended Fig1.

Minor comments:

1. The number of differentially expressed genes in NB subtypes (in Fig4b) is affected by the number of cells, is the conclusion based on this reliable?
2. There is a problem with the layout of the supplementary tables, leading to hundreds of pages in the merged PDF, please check.
3. Is there any cell-cell communication analysis in the ctrl group? Are there differences in cell-cell communication between the ctrl and BM metastases group?
4. The author used many schematic diagrams to help the reader understand the study, which is vivid and adds to the readability of the article, but authors are supposed to enhance the clarity of the diagrams, adjust the position of the diagrams (such as the figure on the left of Fig3d), and add appropriate legends to describe them.

Reviewer #4 (Remarks to the Author): expertise in proteomics

Reviewer Comments:

In the present manuscript, the Fetahu et al. conducted single-cell transcriptomics (scRNA-seq) and epigenomic profiling (scATAC-seq) to investigate the BM niche in neuroblastoma. By comparing 11 NB samples with five age matched and metastasis-free BM (controls), they presented an integrated map of epigenetic and transcriptomic effects of BM metastases at single cell resolution. They further demonstrated cellular plasticity of NB tumor cells is conserved upon metastasis and tumor cell type composition is NB subtype-dependent. Indeed, it is quite interesting work, I have my specific comments as below:

1. The paper needs English proofreading. Most of the time the language can be understood but the scientific interpretation can be dubious, many places it somewhat unclear what is meant by a sentence. In that line, there are sometimes discrepancies/unclearities of what is described in the results section and what is presented in figures and/or figure texts. Please have someone carefully read through the paper checking that results are in line with what the figure says. That the results and figure text clearly describe the data in the figure so it can easily be understood.
2. The methodologies utilized need substantial details, specifically how did the scATAC-seq conducted.
3. The data as presented was considered descriptive and correlative in nature. More in-depth mechanistic validation would be required.
4. One of the major findings of the manuscript is that the author assumed NB cells could rewire specifically monocytes via cell-cell interaction signal to the BM microenvironment. Although it is an intriguing finding, the author should provide further solid evidence, like functional experiments utilizing primary cell cultures.
5. The comparison shown on Figure 4a was between NB metastases and controls, while the comparison in "studies of primary tumor site have found increased..." was between primary tumors with low-/high- risk. Thus, the conclusion "suggesting variation in cell type abundance between the primary and the metastatic site." is particularly problematic.
6. The author should compare their work with other published single cell studies (Cancer Cell, PMID: 32946775; Sci Adv. PMID: 33547074).
7. All data sets should be submitted for public access.

Minor comments.

1. How does Figure 1b-d reflect "we identified a cluster of cells classified as neurons, expressing key NB markers, which were absent in control samples?" It seems that the controls are not presented in the Figure 1b-d.
2. Is the "canonical marker gene" in Figure 1d previously reported? Citations are necessary here.
3. I'm not sure what a "the latter reported in [3]" means, please clarify the findings clearly.
4. Multiple comparisons are done to using the scRNA-seq data and it is unclear which statistical test method of the data that are used in the analyses. e.g., Extended Data Fig. 2a and Figure 4a.

Proteomics comments:

1. As I mentioned in the comments, the methodologies of the manuscript lack substantial details. Specifically, for MS spectrum analysis, they simply referred previous published work without describing the equipment and parameters they utilized.
2. They did not describe how they normalized proteomic data, neither did they provide any information on the quality control of proteomic data.
3. They did not provide their mass spectrum raw data (this actually very critical).
4. Since the proteomic data include those leveraged from previous published work and their newly generated data, the author should describe how they combined those data for integrative analysis.

POINT-BY-POINT RESPONSE TO THE REVIEWERS' COMMENTS

Key: *Reviewers' comments in black italic; Authors' responses in blue.*

Reviewer #1 (Remarks to the Author): expertise in single cell sequencing

Fetahu and colleagues have generated a substantial amount of information at single cell level of human BM metastases in NB. They have performed a thorough analysis determining cell type landscape, NB subtype composition, cell to cell communication between cell types and further analyzed the different myeloid sets. Besides, they correlated this transcriptome information with epigenetics.

The data is really valuable and the analysis performed well detailed and justified. I'm only concerned about the following points:

Integration: authors have made use of batchelor (FastMNN) to perform integration of the samples. Tran et al, 2020 [1] benchmarked these kinds of integration methods, showing better performance from others (LIGER, Harmony,...) than FastMNN. The authors do not justify why they used this method. Could the authors discuss the effect of the analysis on integrating with different methods?

We thank the reviewer for raising this insightful point. For the current work, we used as a guideline a recent study that performed a benchmarking of various integration approaches, which showed that fastMNN was among the top-performing integration methods [1]. In line with these findings, fastMNN outperformed other data integration methods, such as Seurat's canonical correlation analysis (CCA) and reciprocal PCA, and clustering on network of samples (conos) when applied to our dataset as well. Notably, fastMNN was the only method that retained NB cells separated from microenvironment cells (Fig. 1b in the main manuscript). By contrast, integration results from other methods were less convincing, as exemplified by Seurat's CCA method (Figure R1), where the tumor cells were mainly found in two clusters that also contained significant fractions of other cell types, which hampered downstream analyses that required well-defined clusters. Overall, cell types were not clearly separated by this approach. In summary, we suspect that integration methods other than fastMNN partially failed in conserving the biological variation (cell type differences) present in our dataset.

Figure R1. UMAP of the overall dataset after integration with Seurat’s CCA method. Neuroblastoma cells are mainly found in clusters 4 and 10.

There is a need for more references. E.g: the authors claimed that "[...] we identified a cluster of cells classified as neurons, expressing key NB markers [...]". Where did the authors obtain these markers?

We thank the reviewer for the thoughtful comments and apologize for the oversight from our side. We have now clarified in greater details the source of the reference markers that were employed to identify the NB cell population throughout the results section, too. The newly added information has been added on page 4, lines 104-106.

FACS Ext fig 1B. How did the authors obtain the NB cells in the FACS? GD2-FITC MARKER? Where are the source data numbers for this fig?

We thank the reviewer for this critical comment. The source data numbers for the NB cells presented in Extended Data Fig. 1b were included in the Extended Data Table 1 during the initial submission. Briefly, we identified the NB cells by staining for the following markers: GD2 and L1CAM. In addition, we stained for CD45, a marker for various hematopoietic cells and CD34, a marker for human hematopoietic stem/progenitor cells (HSPCs). To discriminate for the NB cells only, we gated for cells that were CD45- and positive for both GD2 and L1CAM markers (as shown in the table below). We have added the gating strategy for these FACS data in in the manuscript on Extended Data Figure 1b whereas the source data numbers for the tumor cells only are listed in Extended Data Table 1, as well a more detailed description in the results section on page 4, lines 108-111.

Extended Data Table R1. FACS quantification of NB cells in the BM marrow samples							
Group	alive	events			% of alive		
		tumor (DTC)	hematopoietic	HSPCs	tumor (DTC)	hematopoietic	HSPCs
		CD45-GD2+L1CAM+	CD45+	CD45+CD34+	CD45-GD2+L1CAM+	CD45+	CD45+CD34+
C1	11880	0	10980	437	0,0	92,4	3,7
C2	17940	1	17496	1829	0,0	97,5	10,2
C3	11387	0	10750	400	0,0	94,4	3,5
C4	17986	0	16589	514	0,0	92,2	2,9
C5	16452	0	15342	309	0,0	93,3	1,9
M1	7973	3020	4915	18	37,9	61,6	0,2
M2	6599	3546	2250	18	53,7	34,1	0,3
M3	6560	2704	3511	245	41,2	53,5	3,7
M4	12975	7308	5008	39	56,3	38,6	0,3
A1	9227	302	8521	76	3,3	92,3	0,8
A2	9452	282	8711	92	3,0	92,2	1,0
S1	2246	20	1490	94	0,9	66,3	4,2
S2	7408	12	7256	300	0,2	97,9	4,0
S3	12959	1990	9841	257	15,4	75,9	2,0
S4	10621	412	10129	118	3,9	95,4	1,1
S5	17282	410	16816	76	2,4	97,3	0,4

Extended Fig1c,d: Did the authors normalize data from Fig1e with the same cells? This is what it looks like from the description. If this is the case is normal that no variations were detected. Authors should take other healthy bone marrow cells to call for variations.

We thank the reviewer for the comment. In the current setting, we employed InferCNV, which predicts copy number alterations by comparing expression levels of genes between tumor cells to normal (i.e., non-malignant) reference cells across the genome. To this end, we labeled the non-NB cells as normal reference cells. The baseline expression in normal cells is then subtracted from both the tumor cells as well as the normal cells, which yields the residual expression values shown in Extended Fig. 1d-e. On the other hand, if the cells classified as normal also contained tumor cells, these would appear in Extended Figure 1d (where we present residual expression of cells we labeled as normal) as clusters of cells with distinct CNV patterns. However, we detect no such signal in the heatmap of normal cells (Extended Figure 1e), with the exception of few outliers in individual cells (and not cell clusters). In contrast, we detected clear and expected signals in the heatmap of tumor cells (Extended Figure 1d). Thus, we conclude that the CNV patterns in these figures confirm our classification of cells as malignant versus non-malignant. We have added more clarity in the methods section regarding this point to prevent any ambiguity for the readers (subsection of the methods - copy number variation calling, page 20, lines 546-553).

The authors show greater expression of noradrenergic markers than the rest. Can the authors test this enrichment? GSEA?

We are grateful to the reviewer for this suggestion. We have now performed gene set enrichment analysis (GSEA) on the said marker set. This in turn, confirmed that adrenergic and noradrenergic signatures were strongly enriched in neuroblastoma cells (Figure R2).

Figure R2. Gene set enrichment analysis of neuroblastoma gene signatures in the neuroblastoma cell cluster.

Authors claim: "The differences between control and NB subtypes [...] were most significant in CD14+CD16- monocytes [...]". The authors show UMAP expression of some of these genes between control and NB subtypes. It would be necessary to quantify and test these differences.

We thank the reviewer for the comments. We have now added more layers of evidence to support the initial data. We now show that CD163, one of markers identified in Extended Data Fig 5d, is upregulated in co-cultured NB cells with PBMCs compared to control, and this specifically occurs in the CD14+CD16+ population as determined by flow cytometry (Extended Data Fig. 5e). Next, we have added a new subfigure where we show that CXCL2, TIMP1, and EREG as significantly upregulated in NB patient samples compared to control (Extended Data Fig. 5e). Finally, by integrating our scATAC-seq and scRNA-seq data we present network analysis where show a link between differentially expressed genes and chromatin accessibility, including for TIMP1 and EREG. These new data have not been presented in Extended Data Fig. 8.

Regarding characterization of the myeloid component within the BM compartment, there are an important amount o myeloid cells non-characterized using the reference-based approach. Could the authors elaborate on the upregulated genes in this cluster?

We thank the reviewer for the insightful comments. As correctly noted by the reviewer, there is a population of cells, classified as "other," which we did not study further. Our rationale was based on the fact that indeed these "other" cells presented in Extended Figure 5b almost exclusively originated from two patients in the control group (Figure R3). Thus, we were not able to compare these cells with the cells derived from non-control patients. This in turn, would

prevent us to generalize our findings across our various patient cohorts, which let us to not further investigate this cell population in more detail.

Figure R3. Contribution of individual samples to the composition of myeloid subclusters.

Authors claim: "[...] loss of gene expression for MYC and E2F targets was associated with predominately closed chromatin". According to Fig5B, these could be for M and S tumors, but not for A. Could the authors correlate these associations and test them? Additionally, switch the figures in Fig5B to place them in accordance with the text.

We thank the reviewer for the important suggestions. As the reviewer rightfully noted, for the M and A NB tumors, the downregulated genes are physically located in the quadrant that is marked with closed chromatin. We now performed correlation analysis, which is shown below along with the correlation coefficients and associated p-values. We observe no correlation or at best a weak correlation for E2F and MYC targets, in the case of NB subgroup S and A, respectively (Figure 4R). The lack of correlation of chromatin accessibility can be attributed to various reasons: 1) location of open chromatin peaks (proximal or distal), 2) degree of differences in peak levels in open chromatin between controls and NB samples. For instance, if we focus on point#1 listed here, we know that there are several layers to chromatin regulation, without even teasing the topic of histone modifications. For instance, a lack of DNA methylation 5mC at promoter regions is generally linked with active genes, but the opposite is observed for lack of 5mC methylation in gene bodies. Moreover, diving deeper into the different layers of DNA methylation, such as 5hmC, which are mainly located on gene bodies of actively transcribed genes, adds another layer of complexity, without forgetting 5fC, 5caC, which also play different roles in gene expression. Therefore, to fully rely on such correlative approaches between RNA-seq and ATAC-seq data from several layers need to be accounted for us to be able to generate reliable and meaningful conclusions. Finally, here we have only included the genes that were identified in the 4 main pathways and have not performed an unbiased correlation analysis between the entire of scRNA-seq and sc-ATAC-seq data sets, which will likely result in a different picture.

Figure R4. Scatterplot and correlation analysis of scRNA-seq and scATAC-seq log fold changes for genes in E2F pathways and MYC targets.

Authors have focused their attention on the myeloid cells, but they also found out that B cells are enriched in inflammatory pathways and depleted. Why the authors have not further investigated changes in the lymphoid compartment, both B and T cells? This seems to have an important impact on NB biology [2]

We thank the reviewer for this excellent question. While this might be on its own an exciting finding, it was however, not within the scope of the current study. We are working to address it in our ongoing studies, which will ultimately aid in better understanding of the interactions of NB cells with other BM microenvironment cell communities beyond myeloid cells.

Github link was not accessible to me.

We apologize for the inconvenience caused to the reviewer. At the time of the initial submission the GitHub repository was private. It will now be made public upon submission of the revised manuscript.

[1] Tran, H.T.N., Ang, K.S., Chevrier, M. et al. A benchmark of batch-effect correction methods for single-cell RNA sequencing data. *Genome Biol* 21, 12 (2020).

<https://doi.org/10.1186/s13059-019-1850-9>

[2] Verhoeven BM, Mei S, Olsen TK, Gustafsson K, Valind A, Lindström A, Gisselsson D, Fard SS, Hagerling C, Kharchenko PV, Kogner P, Johnsen JI, Baryawno N. The immune cell atlas of human neuroblastoma. *Cell Rep Med*. 2022 Jun 21;3(6):100657. doi:

10.1016/j.xcrm.2022.100657. Epub 2022 Jun 9. PMID: 35688160; PMCID: PMC9245004.

Reviewer #2 (Remarks to the Author): expert in brain cancer bone metastasis

The Letter to Nature communications entitled “Dissecting the cellular architecture of neuroblastoma bone marrow metastasis using single-cell transcriptomics and epigenomics unravels the role of monocytes at the metastatic niche” by Fetahu et al. presents a very interesting and informative investigation into the metastatic niche of high risk neuroblastoma using scRNA and scATAC seq approaches. They utilize patient material from 11 different diagnostic cases that represent MYCN amplified and non-amplified versions of neuroblastoma and did an excellent job presenting the rationale for their approach to analyze

these samples.

Overall, this is a very well executed work and I would recommend publishing without major revisions. The conclusions of the manuscript are consistent with current concepts of neuroblastoma biology and shed particular insight into the monocyte/macrophage components of bone marrow interacting with metastatic cancer cells. These results are noteworthy as pharmaceutical and targeted approaches to interfering with tumor suppressive immune cells are rapidly being developed and are expected to help improve the outcome for difficult to treat metastatic solid tumors such as Neuroblastoma.

Two points:

1) The MYCN amplified tumors have a much higher percentage of tumor involvement in the bone marrow compared to the atrx and sporadic subtypes. – How this may confound the sequencing data analysis should be directly addressed (Facs %37-56 versus 2-25% from table 1). It is not clear to me how this increased in tumor cell representation was accounted for in the data analysis.

We thank the reviewer for this excellent point. In the current study, the tumor infiltration rate has been included as a fixed effect in the binomial mixed models, which was fitted for differential gene expression analysis. Thereby, results of this analysis have been corrected for the effects of different tumor infiltration rates. Although we considered of adding infiltration rate as a covariate as the best approach, we also performed the differential expression analysis without this covariate and achieved very similar results. We have now extended our description of the model fitted in the methods section in order to clarify this point (methods section – Differential gene expression and gene set enrichment analysis in scRNA-seq data) The new information can be found on page 22, lines 593-598.

2) The authors highlight the MK and MIF pathways based on bioinformatic assessment of ligand/receptor expression across the various cell types (figure 3). To this reviewer the inferred interactions are interesting but the validation of this type of gene expression at the protein level and individual receptor ligand pair level is lacking. Since we know that many receptors or pseudoreceptors are expressed that do not signal at all in various tumor subpopulations, I find the data in figure 3 weaker than the rest of the manuscript. However, it remains to be seen how useful this type of bioinformatic inference is.

The reviewer raises an excellent point. To address the reviewer's comments, we had initially added the proteomics data from NB cell lines and tumors and respective controls (ganglioneuroma and hMSCs), where we show that these ligands exhibit higher expression in NB tumors and cell lines compared to controls (Figs. 3h-i). In addition, we now established a new *in vitro* model system to study the interactions between NB and immune cells. We co-cultured NB cells with PMBCs and each cell type alone, and used ELISA and flow cytometry as readouts. We determined the secretion of both ligands (MIF and MDK) as well as cell surface expression of their corresponding receptors (CD44, CD74, CXCR4, LRP1, NCL) in various monocyte populations, where we show high expression of these receptors in monocytes. On the other hand, MIF and MDK were secreted by tumors cells and their

secreted levels were even induced in the co-culturing settings, suggesting their interactions and potential feedback loops. These new data have now been included in Figs. 3j-k and Figs. 4h-j, Extended Data Fig. 5e, along with descriptions throughout the manuscript (pages 6-7, lines 157-172 and page 8, lines 208-217).

3) While the published data on core regulatory networks and TF regulation in Neuroblastoma is sighted, it would be helpful if the discussion expanded on how the data here reflects the previous work (which was mostly in cell lines, and I believe did not include metastatic samples).

We thank the reviewer for this excellent comment. We employed different approaches to ensure that we did not overlook the presence of any different cell types expressing the transcriptomic signatures described in [2, 3], however, we did not detect any NB cells expressing NC-like or mesenchymal signatures, albeit they were present in other non-NB cell types. However, while this manuscript was under revision a study [4] showed the presence of mesenchymal cells in patients. As the reviewer mentioned, it is possible that due to these signatures having been initially identified in cell lines could be one of the reasons that in our cohort we do not detect cells with NC-like or mesenchymal signatures. This could also be attributed to the differences in single cell technologies and derived data sets. As the reviewer has suggested, we have now added this information in the discussion to add more clarity (page 11, lines 287-290).

One comment – while MYCN amplified and ATRX mutant tumors appear to have defined drivers, it seems premature to term all the rest of Neuroblastoma as Sporadic if it lacks a MYCN or ATRX mutations.

We agree with the reviewer that the term “sporadic” might be misleading and now explain the use of the term in the introduction (page 2, lines 59-60) as “...but show that a subgroup of metastatic NB is rather defined by large segmental chromosomal aberrations [13] (herein referred to as sporadic).”

• Are there any flaws in the data analysis, interpretation and conclusions? Do these prohibit publication or require revision? I do not find any obvious issues with the data in this manuscript

We thank the reviewer for the comment and appreciate the feedback.

- Is the methodology sound? although I am not expert in the integration of ATAC and RNA seq sc data sets, the presentation is very clear and methodology is sound.*
- Does the work meet the expected standards in your field? Yes*
- Is there enough detail provided in the methods for the work to be reproduced? Yes*

We thank the reviewer for the comment and appreciate the feedback

Overall, this work sheds additional light on tumor/immune cell interactions utilizing well defined patient samples that are representative of the typical spectrum of Neuroblastoma tumors. In addition, they highlight novel and potentially targetable immune subpopulations (pending validation).

Again, we thank the reviewer for the comments and we appreciate the feedback, which led to substantial improvements of the manuscript.

Reviewer #3 (Remarks to the Author): expert in scRNA-seq and ATAC-seq bioinformatics techniques

In this study, Irfete et al. performed scRNA-seq and scATAC-seq of cells in bone marrow from 5 samples of benign tumors without BM metastases and 11 samples across metastatic subtypes. Firstly, the data of the first single cell transcriptome and paired chromatin accessibility atlas of human BM metastases from NB are quite precious. Secondly, the downstream analysis covers many interesting layers, including CNV analysis, cell-cell communication, and so on. Thirdly, some experiments, such as SNP array and protein quantitation, in addition to sequencing data, are included for validation. Overall, Irfete et al. did a valuable study, but the following points could still be improved.

Major comments:

1. The introduction to the background is a bit confusing and vague, leaving no idea what biological problem the author is trying to solve. An interesting question is what determines or influences the BM metastases from NB? What patients will and will not metastasize? Are there public sequencing data of NB primary tumors with and without metastasis available for analysis? Is metastasis determined by the primary site (the cancer cell is too strong) or by the metastasis site (the normal cells in metastasis site are too weak)?

We thank the reviewer for this excellent suggestion. We agree with the reviewer that during the initial submission phases, the manuscript was too condensed, due to space constrains, which at times led to being vague and confusing. Now, we have transformed the manuscript from a “Letter” format to an “Article” format, and as such it has been quite transformed by the addition of the introduction and discussion sections, which we hope it will aid in clarity and eliminate any ambiguity.

2. The authors only showed the basic quality description from cellranger in extended data table, and more quality control analyses of scATAC-seq are needed. For example, the fragment size distribution and TSS enrichment score are recommended.

We thank the reviewer for the helpful suggestions. To ensure the quality of the scATACseq data we conducted rigorous quality control analyses of both scRNA-seq and scATAC-seq data. We provide key measures, which are reported in the Figure R5 (see below), including the nucleosome signal score, the TSS enrichment score, and the fragment size distribution.

which ensure less noisy nucleosome free fragments. We have now (i) revised the description of the filtering criteria in the methods, (ii) revised the reporting of QC measures in Extended Data Supplementary Table 10, and (iii) included a plot of QC measures in Extended Data Fig. 9, as well as in methods section on page 23, lines 614-619.

Figure R5. Summary of quality control reports for scACTA-seq data.

3. Since the authors have identified some interesting molecules or pathways in BM metastases from NB, such as *VEGF* and *EREG*, the regulatory network can be further constructed from scRNA-seq and scATAC-seq. In addition, as the enriched motifs are listed in Fig5d, the author could find the target genes of these TFs by inferring the regulatory network from scRNA-seq and checking the peak-to-gene linkages from scATAC-seq, which may link the two types of sequencing and provide insights in BM metastasis.

We are grateful to the reviewers for the suggestion, which has greatly improved our analyses. Here, we have now integrated scRNA-seq and scATAC-seq datasets using a graph-based integration algorithm [5]. Next, we inferred a regulatory network, which highlighted the connections from the TFs identified by motif analysis on scATAC-seq data to the

differentially expressed genes identified from the scRNA-seq data. Indeed, this analysis now links TF modules to EREG and other target genes involved in monocyte polarization and tumor metastasis, such as IL10, CXCL2 and TIMP-1. These data have now been presented in Extended Data Fig. 8, and highlighted in results (page 10, lines 254-256) and methods sections (pages 25, lines 663-671).

4. The authors are supposed to integrate scATAC-seq and scRNA-seq, which may lead to interesting new clues. There are many tools for integrating multi-omic sequencing. For instance, the authors could annotate clusters of scATAC-seq using scRNA-seq and identify predicted cis-regulatory elements. This may also explain to some extent which important changes are caused by epigenetics and which are caused by other influences such as CNV.

We thank the reviewer for these valuable suggestions. Here, we have now integrated the scATAC-seq and scRNA-seq data sets using GLUE (Graph Linked Unified Embeddings [5]), which is specifically designed for unpaired single-cell datasets. GLUE uses peak to gene correlations to connect scATAC-seq and scRNA-seq datasets. With this integration, we were able to (i) validate our cell type assignments, which were originally performed separately on the two data modalities, but perfectly overlapped upon integration (Extended Data Fig. 9f-g), and (ii) infer a network connecting TFs and their targets (Extended Data Fig. 8a-c), and methods sections (pages 25, lines 663-671).

5. There is some ambiguity in the description of whether all cells or certain types of cells were used in a particular analysis, such as CNV inferred from scRNA-seq in Fig1e and ATAC-seq signal in Fig5c. And why not view them separately by cell type?

We apologize to the reviewer for the ambiguity from our side. Fig. 1e shows only NB cells, whereas Fig. 5 shows only the myeloid cell component. We have updated the figure legends to reflect this point clearly, and we hope it has added sufficient detail to prevent any ambiguity.

6. Many of the pictures are not clear enough, such as Fig1a and schematic illustration in other figures. And some colors in figure are too similar to distinguish, such as colors representing sample C3, M3 and S3 in extended Fig1.

We thank the reviewer for the helpful suggestions. We have amended the figures throughout the manuscript to increase the clarity of the presented data. Specifically, regarding Extended Fig. 1, we agree with the reviewer that some of the colors denoting individual patients are rather similar. Unfortunately, it was challenging to find 16 colors that are clearly separable. However, our rationale for using these colors was as follows: (i) The main point of Extended Data Fig. 1a is to show that before integration, cells from individual patients form distinct clusters (which is visible even though the patients are not discernible), while integration yields larger clusters that comprise cells of many patients. (ii) Similarly, Extended Data Fig. 1e shows that cells do not cluster by patient (again, this is visible although patients are not discernible). Finally, during this revision, we have also uploaded all the figures as images to

ensure high resolution quality for reviewers, hoping to further resolve this issue.

Minor comments:

1. *The number of differentially expressed genes in NB subtypes (in Fig4b) is affected by the number of cells, is the conclusion based on this reliable?*

We thank the reviewer for highlighting this point. We agree with the reviewer that this conclusion is not well substantiated, and we have now removed that conclusion, and amended that section of the results accordingly.

2. *There is a problem with the layout of the supplementary tables, leading to hundreds of pages in the merged PDF, please check.*

We thank the reviewer for the comment. We noticed this issue, too, however, it appears to be a formatting-related problem in the online submission portal of the journal, especially when large excel tables are included. Unfortunately, in the current stage there is no way to circumvent it, as once uploaded, all the files again will be automatically merged into a single PDF document. Moreover, some of the excel files are extensive and simply cannot be fitted into an A4 format.

3. *Is there any cell-cell communication analysis in the ctrl group? Are there differences in cell-cell communication between the ctrl and BM metastases group?*

We thank the reviewer for the helpful suggestions. While this might be on its own be an exciting finding, it was, however, not within the scope of the current study. In the current study, we focused on the interactions between the tumor and the microenvironment cells. Since control patients harbor no NB cells, this analysis cannot be performed in control patients. However, it would tremendously valuable to study how the microenvironment cellular communities interact with each other in the absence of tumor cells.

4. *The author used many schematic diagrams to help the reader understand the study, which is vivid and adds to the readability of the article, but authors are supposed to enhance the clarity of the diagrams, adjust the position of the diagrams (such as the figure on the left of Fig3d), and add appropriate legends to describe them.*

We thank the reviewer for the helpful suggestions. We have amended the figures throughout the manuscript to increase the clarity of the presented data.

Reviewer #4 (Remarks to the Author): expertise in proteomics

Reviewer Comments:

In the present manuscript, the Fetahu et al. conducted single-cell transcriptomics (scRNA-seq) and epigenomic profiling (scATAC-seq) to investigate the BM niche in neuroblastoma.

By comparing 11 NB samples with five age matched and metastasis-free BM (controls), they presented an integrated map of epigenetic and transcriptomic effects of BM metastases at single cell resolution. They further demonstrated cellular plasticity of NB tumor cells is conserved upon metastasis and tumor cell type composition is NB subtype-dependent. Indeed, it is quite interesting work, I have my specific comments as below:

1. The paper needs English proofreading. Most of the time the language can be understood but the scientific interpretation can be dubious, many places it somewhat unclear what is meant by a sentence. In that line, there are sometimes discrepancies/unclarities of what is described in the results section and what is presented in figures and/or figure texts. Please have someone carefully read through the paper checking that results are in line with what the figure says. That the results and figure text clearly describe the data in the figure so it can easily be understood.

We thank the reviewer for the comment. We agree with the reviewer that the manuscript during the initial submission was too condensed, which at times due to space constraints led to vagueness and ambiguity. Now, we have transformed the manuscript from a “Letter” format to an “Article” format, and as such with the addition of introduction and discussion sections we hope it will increase the clarity of the manuscript. In addition, two native English speakers have edited the manuscript for language clarity and correctness.

2. The methodologies utilized need substantial details, specifically how did the scATAC-seq conducted.

We apologize for the lack of clarity. We have now revised the methods section accordingly and have added more information that should provide better clarity (methods section, page 23, lines 613-618). Additionally, we have now integrated the scATAC-seq and scRNA-seq data sets using GLUE (Graph Linked Unified Embeddings [5], which is specifically designed for unpaired single-cell datasets. GLUE uses peak to gene correlations to connect scATAC-seq and scRNA-seq datasets. With this integration, we were able to (i) validate our cell type assignments, which were originally performed separately on the two data modalities, but perfectly overlapped upon integration (Extended Data Fig. 9), and (ii) infer a network connecting TFs and their targets (Extended Data Fig.8), and methods sections on page 25, lines 663-671.

3. The data as presented was considered descriptive and correlative in nature. More in-depth mechanistic validation would be required.

We thank the reviewer for highlighting this issue. We have provided extensive new data to support and validate the findings that were reported in the first submission of this manuscript. Specifically, we now established a new *in vitro* model system to study NB – immune cell interactions between the identified ligand/receptor pairs. We co-cultured NB cells with PMBCs and each cell type alone, and used ELISA and flow cytometry as readouts. We

determined the expression of secreted ligands: MIF and MDK, showing high levels of secretion by tumors cells and their secreted levels were even higher in the co-culturing settings. Moreover, we tested for the expression of their corresponding receptors: CD44, CD74, CXCR4, and LRP1, NCL, respectively, in various myeloid populations, which showed high expression of these receptors in monocytes. These new data have now been included in Figs. 3j-k and Figs. 4h-j, Extended Data Fig. 5e, along with descriptions throughout the manuscript (pages 6-7, lines 157-172 and page 8, lines 208-217).

4. One of the major findings of the manuscript is that the author assumed NB cells could rewire specifically monocytes via cell-cell interaction signal to the BM microenvironment. Although it is an intriguing finding, the author should provide further solid evidence, like functional experiments utilizing primary cell cultures.

We thank the reviewer for this excellent question. In addition to validating our findings regarding ligand/receptor interactions in our co-culturing model as described above in point#3 and Figures 3&4, we also show that monocytes co-cultured directly with NB cells exhibit high secretion of inflammatory M1 cytokines, IFN γ , TNF α and IL-1 β and M2 cytokines, IL-10 and TGF β as well as a CD163+CD86dim phenotype, in line with M2-like cultures (stimulated with IL-4 and IL-10). This was in stark contrast to M1-like cultures (stimulated with IFN γ and LPS), which display predominantly a CD163dimCD86high phenotype. Furthermore, monocytes co-cultured through a trans-well with NB cells display an increase in the fraction of CD14+CD16+ cells and a slightly different phenotype (higher expression of CD163), however, exhibit an M2-like phenotype as well. Finally, expression levels of MHC class I&II markers show that overall co-cultured NB cells with PBMCs display similar levels as under M2-like conditions. This corroborates our initial findings and suggests that monocytes in the presence of NB cells, undergo a rewiring process where they exhibit an inflammatory activated M1 and M2-like phenotype (Figs 3-4, pages 6-7, lines 157-172 and page 8, lines 208-217).

5. The comparison shown on Figure 4a was between NB metastases and controls, while the comparison in “studies of primary tumor site have found increased...” was between primary tumors with low-/high- risk. Thus, the conclusion “suggesting variation in cell type abundance between the primary and the metastatic site.” is particularly problematic.

We thank the reviewer for pointing this out. While the primary NB tumors have been better characterized, the metastatic niche of NB, the bone marrow, especially at single cell resolution has not been described yet. The statement above was meant to convey the following: that in the metastatic NB niche we observed the opposite of what has been described in primary tumors with regard to T, NK, and B cells, indicating that the primary and metastatic tumor microenvironment exhibit major differences in their immune components. However, in the current we have removed that statement and amended the text to avoid any ambiguity, specifically by adding more information in the introduction (page 3, lines 70-74).

6. *The author should compare their work with other published single cell studies (Cancer Cell, PMID: 32946775; Sci Adv. PMID: 33547074).*

We thank the reviewer for the comment. The mentioned references were already included in the original submission in the introductory paragraph and the results section. However, now we have emphasized them further in the discussion section, too. With regards to data analysis, we focused on the bone marrow metastasis and integrated the data set by Dong et al. [6] for comparison with primary tumors in Fig. 2c and Extended Data Fig. 2e. The motivation to include the dataset by Dong and colleagues [6] in our analysis was because it presents a homogeneous cohort of patients with high-risk neuroblastoma, where the tumors were obtained at diagnosis. On the other hand, the study by Kildisiute et al., [7] describes a heterogeneous cohort of benign and high-risk forms of neuroblastoma, primary tumors and metastasis, which were obtained before or even after treatment.

7. *All data sets should be submitted for public access.*

We thank the reviewer for the helpful comment and fully agree with it. The data analyses code will be publicly available from GitHub upon submission of the revised manuscript. Raw sequencing data are already publicly available from EGAS00001006106 upon request, in line with the General Data Protection Regulation. Count matrices have been deposited in the Gene Expression Omnibus (SuperSeries GSE216176) and will be publicly released upon publication. Currently, they are accessible via the reviewer token ‘ynovcewcvvadxmf’. Proteomics data will be available from PRIDE (PXD036979 and PXD036972).

Minor comments.

1. *How does Figure 1b-d reflect “we identified a cluster of cells classified as neurons, expressing key NB markers, which were absent in control samples?” It seems that the controls are not presented in the Figure 1b-d.*

We apologize for the vague description and ambiguity. Figs. 1b-d include control patients. Extended Data Fig. 1b confirms that control patients do not contain any tumor cells (where the tumor infiltration rate is zero as determined by FACS, Extended Data Fig. 1b and Extended Data Table 1). We have explained in greater detail this part in the current draft of the manuscript to provide more clarity (page 4, lines 104-106, and page 4, lines 108-111).

2. *Is the "canonical marker gene" in Figure 1d previously reported? Citations are necessary here.*

We thank the reviewer for the comment. The references for the marker genes are now also included in the appropriate results sections in addition to the methods section (page 4, lines 101-106).

3. *I'm not sure what a “the latter reported in [3]” means, please clarify the findings clearly.*

We thank the reviewer for the comment. We have edited that sentence for clarity, and it now reads as follows ‘Leveraging published scRNA-seq data of primary tumors [17], we show that bone marrow metastases display tumor cell phenotypes comparable to the primary site, which were reported in [18].’ page 5, lines 126-128.

4. Multiple comparisons are done to using the scRNA-seq data and it is unclear which statistical test method of the data that are used in the analyses. e.g., Extended Data Fig. 2a and Figure 4a.

We thank the reviewer for the comment. We have addressed this issue in the methods section, where we have stated that all p-values from multiple tests have been corrected by the Benjamini-Hochberg method, page 22, lines 590-591.

Proteomics comments:

1. As I mentioned in the comments, the methodologies of the manuscript lack substantial details. Specifically, for MS spectrum analysis, they simply referred previous published work without describing the equipment and parameters they utilized.

We thank the reviewer for the comment. In addition to the references where the reported data originate from, wherein the methods were described as well, we have now included in the methods section more details on the equipment and the settings employed for the measurements (page 15, lines 397-412 and page 16, lines 433-447).

2. They did not describe how they normalized proteomic data, neither did they provide any information on the quality control of proteomic data.

We thank the reviewer for these insightful comments. The normalization of the proteomics data is done in two steps. First, a defined amount of protein (20µg) of each sample was used for the enzymatic digestion. Second, the MaxQuant software algorithm normalizes the raw data based on the total ion chromatogram (TIC) within the data set.

Regarding the quality control, four synthetic peptides [Glu1-fibrinopeptide B, EGVNDNEEGFFSAR; M28, TTPAVLDSGDGFLYSK; HK0, VLETKSLYVR; HK1, VLETK(ε-AC)SLYVR] were spiked in each sample prior to LC-MS/MS analyses. These peptides allow us to monitor the retention time stability as well as the mass spectrometer performance regarding peak intensities (page 15, lines 397-412 and page 16, lines 433-447).

3. They did not provide their mass spectrum raw data (this actually very critical).

We thank the reviewer for the helpful comment. During the course of the revision, the proteomic data of primary control and activated human monocytes as well as of M1- and M2-like macrophages, including all raw data and mass spectra have been deposited at the ProteomeXchange Consortium via the PRIDE partner repository with the dataset identifiers:

a. PXD036979, Username: reviewer_pxd036979@ebi.ac.uk, Password: wGKETwhR

b. PXD036972, Username: reviewer_pxd036972@ebi.ac.uk, Password: p9h1m0Op

4. Since the proteomic data include those leveraged from previous published work and their newly generated data, the author should describe how they combined those data for integrative analysis.

As the reviewer correctly noted, different data sets were used in the present study. Moreover, as the reviewer pointed, we are fully aware, too, that a combination of proteomic data sets derived from different instrumental setups and a direct comparison of protein abundances is not valid due to normalization problems. Therefore, we analyzed each data set separately and used it to underly specific results in the present study. We did not compare protein abundances between the different data sets nor draw conclusions from such comparisons since this would not be appropriate as the reviewer already mentioned.

References

1. Luecken, M.D., et al., Benchmarking atlas-level data integration in single-cell genomics. *Nat Methods*, 2022. 19(1): p. 41-50.
2. Boeva, V., et al., Heterogeneity of neuroblastoma cell identity defined by transcriptional circuitries. *Nat Genet*, 2017. 49(9): p. 1408-1413.
3. van Groningen, T., et al., Neuroblastoma is composed of two super-enhancer-associated differentiation states. *Nat Genet*, 2017. 49(8): p. 1261-1266.
4. Yuan, X., et al., Single-cell profiling of peripheral neuroblastic tumors identifies an aggressive transitional state that bridges an adrenergic-mesenchymal trajectory. *Cell Rep*, 2022. 41(1): p. 111455.
5. Cao, Z.J. and G. Gao, Multi-omics single-cell data integration and regulatory inference with graph-linked embedding. *Nat Biotechnol*, 2022. 40(10): p. 1458-1466.
6. Dong, R., et al., Single-Cell Characterization of Malignant Phenotypes and Developmental Trajectories of Adrenal Neuroblastoma. *Cancer Cell*, 2020. 38(5): p. 716-733.e6.
7. Kildisiute, G., et al., Tumor to normal single-cell mRNA comparisons reveal a pan-neuroblastoma cancer cell. *Sci Adv*, 2021. 7(6).

REVIEWERS' COMMENTS

Reviewer #1 (Remarks to the Author):

Fetahu and colleagues have made an important effort on the manuscript and figures according to my questions and the rest of the reviewers. All my questions have been addressed. I especially congratulate the team on the GitHub repository. All the analyses are nicely available and reproducible, along with the original data.

Reviewer #2 (Remarks to the Author):

The revised manuscript by Fetahu et al (Dissecting the cellular architecture of neuroblastoma bone marrow metastasis using single cell transcriptomic...) has been significantly improved and the responses to previous reviews have been adequately addressed.

I only have a few additional comments -

line 122: The statement that 'no NB cells with a mesenchymal or NCC-like signature were discernible' requires additional caveats- and should be modified considering that most single cell analyses of patient samples clearly define a small mesenchymal population. so 'in our sample sets, at the resolution of this study we did not detect mesenchymal subtypes' In addition the very next sentence mentions 'bridge cells' in the MNA bma samples - these are defined as mesenchymal like subpopulations in other studies. Care must be taken not to exclude populations not detected- as mesenchymal subpopulations are only a small fraction of total NB cells sampling errors could lead to missing this population.

The second paragraph of discussion line 273 -289 reflects the same theme- trying to explain why mesenchymal subtypes were not seen in the bone marrow samples. I think the answer is 'we don't know'

could be sampling error, or mes cells don't freeze well, or the tumors analyzed happen to have very low mesenchymal populations that our methods were not sensitive enough to detect... - consider revising this paragraph to limit this discussion as it does not significantly contribute to the main points of the manuscript. previous reports suggest that high risk tumors have small schwannian like precursor and other chromaffin precursor populations that are mesenchymal.

Reviewer #3 (Remarks to the Author):

The authors have expanded on the quality control of scATAC-seq, the integrated analysis of multi-omics by GLUE, and the regulatory network, which addressed my concerns. I would recommend publishing the article and I suggest the authors could further improve from the following points. 1. The methods for gene regulatory network inference are not clear. 2. The Extended Data Fig. 8 takes up three pages, which could be resized to one page.

Reviewer #4 (Remarks to the Author):

The authors have adequately addressed all my comments.